# Perceptions of the appropriate response to norm violation in 57 societies

Norm enforcement may be important for resolving conflicts and promoting cooperation. However, little is known about how preferred responses to norm violations vary across cultures and across domains. In a preregistered study of 57 countries (using convenience samples of 22,863 students and non-students), we measured perceptions of the appropriateness of various responses to a violation of a cooperative norm and to atypical social behaviors. Our findings highlight both cultural universals and cultural variation. We find a universal negative relation between appropriateness ratings of norm violations and appropriateness ratings of responses in the form of confrontation, social ostracism and gossip. Moreover, we find the country variation in the appropriateness of sanctions to be consistent across different norm violations but not across different sanctions. Specifically, in those countries where use of physical confrontation and social ostracism is rated as less appropriate, gossip is rated as more appropriate.

Norms, in the sense of collective ideas about approved and disapproved behavior, exert a powerful influence on how people behave[1]. However, not everyone complies with these norms, which may create dilemmas for those who witness norm-violating behaviors and must decide whether to respond with some kind of sanction. On the one hand, previous work has suggested that norms encouraging informal sanctions are critical to sustaining cooperation and social order in human groups[2–4]. On the other hand, unfettered or inappropriate use of sanctions may threaten social harmony by creating costly conflicts[5,6]. Thus, cooperation and social harmony depend on norms about the use of informal sanctions. Such norms about norm enforcement have been termed metanorms[7]. Despite their importance, surprisingly little is known about how metanorms operate in everyday life, let alone across societies.

Existing research often examines and conceptualizes sanctions in generic terms as a form of punishment that reduces outcomes for another person[8,9]. While parsimonious, this characterization is unlikely to provide a realistic account of how people deal with norm violators in everyday life. To capture this realism, scholars[10,11] have recently proposed three distinct informal sanctions: social ostracism (e.g., individuals or groups actively avoiding someone), gossip (e.g., spreading information about someone's inappropriate behavior), and direct confrontation (e.g., verbal or physical). Although these responses may not always be intended to modify the norm violator's behavior, they can all be viewed as expressions of disapproval that serve to strengthen a given norm. A key reason that potential norm enforcers may prefer one response over another is that responses may differ in the extent to which the sanctioned party becomes aware of being sanctioned. For instance, whereas direct confrontation should be especially effective at making the norm violator aware of why they are being sanctioned and thus change their behavior, gossip should be less likely to evoke direct conflict but can still promote norm compliance by making the norm more salient in the group. Similarly, physical confrontation may be harmful in a way that verbal confrontation is not. And social ostracism may directly harm targets' opportunities whereas gossip may harm targets more indirectly via reputational damage. Prior cross-cultural work has rarely distinguished between forms of sanctions, instead focusing on costly actions that reduce outcomes for another person in economic games[12,13], physical confrontation[14], or unspecified "punishment"[15].

To compare the perceived appropriateness of different forms of sanctions across societies, we studied participants in 57 countries, including 7 African countries, 10 American countries, 18 Asian countries, 21 European countries, and Australia. The study included 10 basic scenarios, mostly drawn from prior studies of norm violations[14,16]. These stimuli covered various domains of norm violations and included both animations and verbal scenarios. One scenario described a violation of a cooperative norm regarding a common resource[14]. Four scenarios described behaviors that were normatively out of place, such as listening to music in headphones at a funeral[16]. Five "meta-violation" scenarios described a potentially overly harsh response to another's behavior, such as someone responding to a verbal insult by physical confrontation. For each of the 10 scenarios, participants rated the appropriateness of the described behavior as well as the appropriateness of four different responses to it: verbal confrontation (making an angry remark to the norm violator), gossip (talking to someone else about the norm violator), social ostracism (making a point of avoiding the norm violator in the future), and non-action (doing nothing), for a total of $10 \times 5 = 50$ ratings.

The study, including the following five key hypotheses, was preregistered with the Open Science Framework (osf.io/qg6xy).

Hypothesis 1: The more appropriate a triggering behavior is perceived to be, the more appropriate it is to respond by doing nothing and the less appropriate it is to respond by using confrontation, social ostracism, or gossip, and this will be consistent across countries. The hypothesized negative relation between the appropriateness ratings of norm violations and the appropriateness rating of a response has previously been reported specifically for verbal confrontation in the United States[17]. But we do not know whether it holds for other forms of sanctions and across cultures. This relationship is important for both conceptual and methodological reasons. Conceptually, only a negative relation would signify that that the sanction is indeed an expression of disapproval. Methodologically, when comparing the perceived appropriateness of a given response across societies, it is important to control for the appropriateness rating of the norm violation as this may differ between societies. The four following hypotheses concern the country variation in the perceived appropriateness of informal sanctions: its consistency across different norm violation domains, its specificity across different forms of sanctions, its relation to variation in the use of informal sanctions, and its relation to variation in other cultural and societal factors.

Hypothesis 2: The country-level variation in the perceived appropriateness of informal sanctions is robust across different domains of norm violations. As metanorms are assumed to serve the function of sustaining cooperation[4], empirical work has focused on norm violations with respect to contributions to, or depletion of, common resources[18–20]. We will refer to this category of norm violations as the "cooperation domain". Importantly, many social norms do not belong to the cooperation domain and may be more mundane, often just stipulating that certain acts are not appropriate in certain contexts[16]. For instance, it tends to be viewed as inappropriate to sleep in a restaurant or to listen to music in headphones at a funeral. For theorizing about the psychology of informal sanctions it is crucial to know whether the cooperative domain of resource dilemmas has a special status or whether the perceived appropriateness of a given response is independent of the domain of the norm violation (holding constant how inappropriate the norm violation is perceived to be). The present research illuminates this issue. Previous work on verbal punishment of uncivil behavior does not indicate any special status of the cooperation domain[17,21], supporting the parsimonious hypothesis that the psychology of norms has a high level of generality that cuts across various domains. Hence, we expected that the appropriateness rating of the norm violation would be leading in influencing evaluations of the appropriateness of the responses, in a manner relatively independent of the domain of the norm violation.

Hypothesis 3: With respect to the specificity of different sanctions we have three competing sub-hypotheses. Comparing across different forms of informal sanctions, the country variation in ratings of their appropriateness will exhibit either (A) consistency, (B) complementarity, or (C) independence. As mentioned earlier, different forms of real-life sanctions seem to be quite distinct, yet much prior work has used a unitary conceptualization of sanctioning as payoff reduction. A unitary conceptualization may be warranted if metanorms vary in the same way for different forms of sanctions, such that some societies view sanctions, in general, as more appropriate than other societies. Another possibility is that all societies employ sanctions but have different preferences for the form they should take, such that a lower appropriateness rating of one sanction is matched by a higher level for another sanction; for instance, it could be that societies prefer either direct confrontation or non-confrontational sanctions such as ostracism and gossip. A final possibility is that different forms of sanctions serve different

purposes and thus have little to do with each other, in which case the appropriateness ratings of different sanctions would be unrelated.

*Hypothesis 4*: In countries where a given sanction is more viewed as appropriate it will also tend to be used more often. This hypothesis constitutes a validation of metanorms. Just as norms influence behavior, metanorms are expected to influence sanctioning behavior.

*Hypothesis 5:* The perceived appropriateness of direct punishment will be higher in countries estimated to have (a) lower on indulgence, (b) higher on power distance, (c) lower on individualism and individual autonomy values, (d) higher on tightness, (e) higher on experienced threat, (f) lower on emancipative moral judgments, (g) higher on pro-violence attitudes, (h) higher on pathogen prevalence, (j) lower on gender equality, and (j) lower on median income. All predictions were preregistered except the last three (which theoretically connect to the other predictions, see below).

To enable examination of Hypothesis 5, our survey includes several culture measures that we aggregate to country level: individual autonomy (valuation of independence and determination over religious faith, and obedience), emancipative moral judgments (how justified it is with homosexuality, divorce, abortion, and suicide), pro-violence attitudes, tightness (pervasiveness of social norms and low tolerance for noncompliance), and perceived threats to society (from disease, conflicts, etc.). We also use data from other sources as follows. We use country measures of indulgence (available for 48 countries in our study), power distance (51 countries), and individualism (51 countries) provided by Hofstede et al.[22]. We use country measures of pathogen prevalence from prior work on the historical prevalence of infectious diseases in different geopolitical regions[23]. We measure national levels of gender equality by the Global Gender Gap Index, which is calculated by the World Economic Forum based on gender gaps in economic participation and opportunity, educational attainment, health and survival, and political empowerment[24]. From Gallup we obtain country measures of median income (50 countries)[25].

Note that the predictions in Hypothesis 5 focus on how the perceived appropriateness of direct punishment (physical and verbal confrontation) will vary across societies; whether the same patterns or the opposite patterns will hold for indirect sanctions like social ostracism and gossip depends on which of the sub-hypotheses of Hypothesis 3 is correct. Our predictions on direct punishment draw on theories of cultural dimensions, societal tightness-looseness, and behavioral responses to the experience of ecological threat. With respect to cultural dimensions, a cross-cultural study found that responding to non-cooperation by physical confrontation was viewed as more appropriate in countries that were characterized by low levels of indulgence (i.e., restrictive of enjoying life and having fun, which a norm violator may be viewed as doing), high levels of power distance (i.e., accepting asymmetry of power, which a punisher may be viewed as wielding), and low levels of individualism (i.e., emphasizing group embeddedness over individual autonomy, which a norm violator may be viewed as expressing)[14]. In line with the role of individualism, another study found verbal confrontation of uncivil behavior to be more normative in less individualistic societies[21]. With respect to tightness-looseness, there is some cross-cultural evidence showing that formal institutions tend to be more punitive in tighter countries[3,16], and our prediction is that this extends to informal sanctioning. Tight societies have generally experienced more collective threat, and direct punishment of deviants may be evolutionarily adaptive in these contexts[26]. By extension, the perceived appropriateness of direct punishment is also expected to be related to the experience of threat. The theory of behavioral immune system similarly traces the origins of cultural differences to ecological threat, especially pathogen prevalence, which is assumed to increase the need for social coordination and thereby lead to cultures with less individualism, greater power distance, and less tolerance of nonconformity[23,27], all of which suggest that pathogen prevalence will also increase the perceived appropriateness of direct punishment. Finally, modernization theory ties the development of cultural values to economic development. Specifically, increased prosperity is assumed to facilitate a shift from traditional values and community discipline to post-material, emancipative values that include a greater emphasis on individual autonomy, gender equality, and emancipative moral judgments[28]. Through a variety of socioeconomic mechanisms, modernization is thought to increase competition and complexity, and reduce interdependence, thereby increasing prioritization of individual freedom, choice, and agency over conformity to the needs or traditions of a society. It is therefore expected to be associated with greater tolerance for a wide range of norm violations and, consequently, a decrease in the perceived appropriateness of punishing them.

In this study of 57 countries we find support for the five hypotheses outlined above. Thus, we find a universal negative relation between appropriateness ratings of norm violations and appropriateness ratings of responses in the form of confrontation, social ostracism, and gossip. The country variation in the appropriateness ratings of sanctions is found to be consistent across different norm violations but not across different sanctions. While the use of confrontation and social ostracism is viewed as less appropriate in more prosperous countries with more emancipative values, the opposite holds for gossip. Our findings thus highlight both cultural universals and cultural variation with respect to beliefs about how norms should be enforced. Perhaps most intriguingly, our findings suggest that responses to norm violators may shift with economic development in a specific way, such that gossip to some extent is used in place of more punitive sanctions, potentially affecting societies' ability to achieve norm compliance.

## Results

All appropriateness ratings were made on a six-point scale from extremely inappropriate (coded 0) to extremely appropriate (coded 5), which were standardized for each respondent to control for response sets. Throughout, numbers in brackets refer to 95% bias-corrected and accelerated confidence intervals based on 1000 bootstrap samples generated by SPSS v. 26.0. The sample size for analyses is $n = 57$ countries unless stated otherwise.

**Hypothesis 1**. As preregistered, we tested Hypothesis 1 in each country by calculating correlations, across the ten scenarios, between country-mean ratings of norm violations and a given response. The boxplots in Fig. 1 illustrate the results, confirming that informal sanctions were essentially universally viewed as less appropriate to use the more appropriate the norm violation was perceived to be. This held for verbal confrontation, M = −0.77, 95% CI [−0.80, −0.75], gossip, M = −0.67 [−0.71, −0.62], and social ostracism, M = −0.39 [−0.44, −0.34]. As predicted, these negative correlations clearly distinguished sanctions from nonaction, for which the correlation was universally positive, M = 0.57 [0.55, 0.60]. Thus, Hypothesis 1 was supported.

**Country measures of metanorms**. Following our preregistration, we calculated country measures of metanorms for each of four responses (verbal confrontation, social ostracism, gossip, nonaction) by using country-mean appropriateness ratings for the

five scenarios in the non-cooperation and out-of-place behavior domains. These were adjusted for variation in ratings of the appropriateness of the underlying norm violations (see "Methods"). Metanorm measures for all countries are reported in Supplementary Table 2 and illustrated on color-coded maps in Supplementary Fig. 1.

In the preregistration we assumed that metanorms for verbal and physical confrontation would be the same, but recent work has shown that they may be viewed quite differently[10]. We therefore separately calculated the country measures of metanorms for physical confrontation by averaging the country-mean appropriateness ratings of two meta-violation scenarios in which physical confrontation was used in two different contexts: against an agent depleting a common resource and against someone insulting a man's mother.

Averaged over all countries, the responses rated as most appropriate were non-action, M = 2.67, 95% CI [2.63, 2.70] and verbal confrontation, M = 2.61 [2.55, 2.66], followed by gossip, M = 2.40 [2.35, 2.46], and then social ostracism, M = 2.02 [1.97, 2.07]. The lowest mean rating was for physical confrontation, M = 1.80 [1.70, 1.90]; however, it is not directly comparable as it is not based on the same set of underlying norm violations.

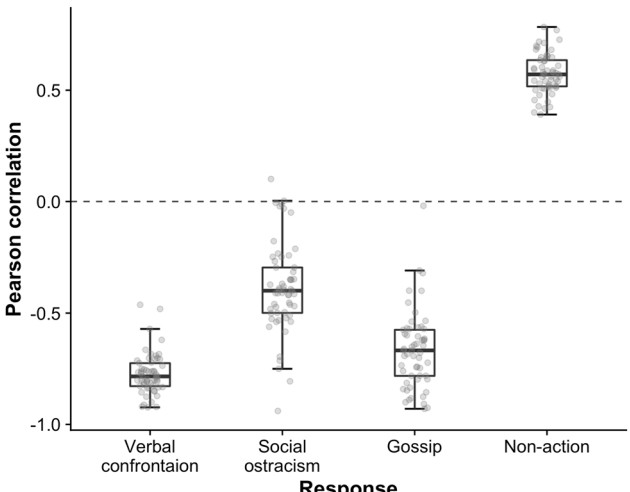

**Fig. 1 Within-country correlations between appropriateness ratings of norm violations and different responses.** The vertical axis refers to the correlation, across n = 10 scenarios, between the (country level) appropriateness ratings of the norm violation and a given response to it. For every response, the corresponding boxplot presents how the value of the within-country correlation varied across n = 57 countries. Verbal confrontation, gossip, and social ostracism almost universally yielded negative correlations, while non-action universally yielded positive correlations. The dashed reference line indicates a zero correlation. The box represents the interquartile range with the dark line indicating the median. The whiskers reach the min and max values in case these are at most 1.5 times the box height outside the interquartile range. Individual data points are overlaid as dots.

Importantly, there was no global consensus on the most appropriate response. Verbal confrontation was rated highest in 26 countries. However, non-action was rated highest in the remaining 31 countries, and in 17 of these countries the highest-rated sanction was gossip. There was even one country (Thailand) where social ostracism was the highest-rated sanction.

As a robustness check of observed country differences, we found that differences in metanorms between cities in the same country, as well as between students and non-students in the same country, tended to be much smaller than the country differences (Supplementary Table 3). Metanorm measures were virtually unchanged in analyses that excluded participants who failed attention or comprehension checks (Supplementary Table 4).

**Hypothesis 2**. As preregistered, we tested the robustness across domains (Hypothesis 2) by examining pairwise correlations of metanorms measures based on different sets of domains. Measures based only on the non-cooperation scenario correlated strongly with the main measures for verbal confrontation, r = 0.79 [0.70, 0.87], social ostracism, r = 0.66 [0.41, 0.81], and gossip, r = 0.92 [0.86, 0.96], and more weakly for non-action, r = 0.29 [0.05, 0.50]. Measures based on all 10 scenarios (i.e., including meta-violations in addition to non-cooperation and out-of-place behavior) correlated very strongly with the main measures, all r > 0.9. For physical confrontation we similarly tested robustness across domains by correlating the main measure with the measure based only on the scenario of physical confrontation against non-cooperation, r = 0.87 [0.81, 0.92]. In an additional, unregistered, analysis we found generally high internal consistency of country-level appropriateness ratings of a given response across different scenarios (Supplementary Table 9). Thus, Hypothesis 2 was supported.

**Hypothesis 3**. Following the preregistration, we analyzed the sanction-specificity of metanorms by calculating pairwise partial correlations of the metanorm measures for different sanctions, controlling for the metanorm measure for non-action (Table 1). When interpreting these correlations, note that they will, to some extent, be artificially lowered due to ratings being standardized. Nonetheless, they present a very clear but complex pattern, simultaneously including all possibilities discussed in Hypothesis 3: consistency, independence, and complementarity. Metanorms for physical confrontation and social ostracism showed a high level of consistency (i.e., were positively correlated), but were largely independent of the metanorm for verbal confrontation. Strikingly, the metanorm for gossip showed strong complementarity (i.e., negative correlations) to the metanorms for all other sanctions.

**Hypothesis 4**. To measure the frequencies by which various informal sanctions are used in different countries, the survey included three items where participants estimated how often they use various responses to someone who does something inappropriate. As preregistered, we tested Hypothesis 3 by calculating the correlations between the country mean for the frequency of

---

**Table 1 Partial correlations between metanorms for different responses.**

| Metanorms | Physical confrontation | Verbal confrontation | Social ostracism |
|---|---|---|---|
| Physical confrontation | | | |
| Verbal confrontation | 0.14 [−0.18, 0.41] | | |
| Social ostracism | 0.56 [0.36, 0.74] | −0.27 [−0.50, −0.01] | |
| Gossip | −0.72 [−0.83, −0.53] | −0.50 [−0.70, −0.28] | −0.57 [−0.71, −0.35] |

*Note.* Partial correlations are based on n = 57 countries, controlling for the metanorm for non-action. 95% confidence intervals are presented in brackets.

**Table 2 Correlations of metanorm measures with other country variables.**

| Predictor | Metanorms for | | | | |
| | Physical confrontation | Verbal confrontation | Social ostracism | Gossip | Non-action |
|---|---|---|---|---|---|
| Indulgence | −0.21 [−0.45, 0.06] | −0.15 [−0.38, 0.09] | −0.24 [−0.43, −0.05] | 0.37 [0.18, 0.54] | −0.04 [−0.32, 0.24] |
| Power distance | 0.69 [0.53, 0.81] | 0.34 [0.15, 0.53] | 0.36 [0.15, .56] | −0.43 [0.64, −0.23] | −0.51 [−0.69, −0.31] |
| Individualism | −0.61 [−0.72, −0.49] | −0.37 [−0.63, −0.04] | −0.37 [−0.55, −0.20] | 0.43 [0.21, 0.64] | 0.48 [0.27, 0.66] |
| Individual autonomy | −0.53 [−0.71, −0.31] | −0.15 [−0.34, 0.04] | −0.24 [−0.51, 0.05] | 0.35 [0.11, 0.60] | 0.39 [0.18, 0.58] |
| Emancipative moral judgments | −0.76 [−0.84, −0.66] | −0.22 [−0.46, .03] | −0.54 [−0.72, −0.33] | 0.54 [0.34, 0.72] | 0.44 [0.21, 0.63] |
| Pro-violence attitudes | 0.35 [0.11, 0.60] | −0.26 [−0.53, 0.09] | 0.39 [0.07, 0.62] | −0.12 [−0.36, 0.18] | −0.12 [−0.42, 0.16] |
| Tightness | 0.50 [0.27, 0.69] | −0.05 [−0.29, 0.17] | 0.37 [0.10, 0.61] | −0.16 [−0.41, 0.09] | −0.30 [−0.48, −0.08] |
| Perceived threat | 0.12 [−0.14, 0.40] | 0.20 [−0.08, 0.43] | 0.04 [−0.18, 0.25] | −0.04 [−0.27, 0.22] | −0.36 [−0.58, −0.11] |
| Pathogen prevalence | 0.47 [0.25, 0.62] | 0.28 [0.06, 0.50] | 0.21 [−0.02, 0.42] | −0.22 [−0.42, −0.02] | −0.45 [−0.66, −0.21] |
| Gender equality | −0.72 [−0.81, −0.61] | −0.11 [−0.30, 0.10] | −0.58 [−0.75, −0.37] | 0.49 [0.26, 0.69] | 0.29 [0.04, 0.50] |
| Median per-capita income | −0.67 [−0.78, −0.54] | −0.36 [−0.61, −0.06] | −0.37 [−0.55, −0.14] | 0.58 [0.40, 0.72] | 0.45 [0.19, 0.65] |

*Note.* Predictors come from different sources: measures of indulgence, power distance, and individualism are from Hofstede et al.[22]; individual autonomy, emancipative moral judgments, pro-violence attitudes, tightness, and perceived threat were measured in the current study; measures of historical pathogen prevalence are from Murray and Schaller[23]; gender equality is measured by the Global Gender Gap Index from World Economic Forum[24]; median per-capita income measures are from Gallup[25]. Correlations are based on $n = 57$ countries, except for indulgence ($n = 48$), power distance ($n = 51$), individualism ($n = 51$), and median income ($n = 50$). 95% confidence intervals are presented in brackets.

use of a response and the metanorm measure for the same response. As predicted, all correlations were positive: the use of "confrontation" correlated with both the physical confrontation metanorm, $r = 0.49$ [0.20, 0.70], and the verbal confrontation metanorm, $r = 0.40$ [0.20, 0.58]; the use of "avoiding" correlated with the metanorm for social ostracism, $r = 0.49$ [0.27, 0.68]; and the use of "talking to others' correlated with the metanorm for gossip, $r = 0.60$ [0.41, 0.75]. Thus, Hypothesis 4 was supported. (All pairwise correlations between metanorms and use of different sanctions are reported in Supplementary Table 5.)

**Hypothesis 5.** As preregistered, we calculated the pairwise correlations between our metanorm measures and the various country measures mentioned in Hypothesis 5 (Table 2). For physical confrontation, all correlations showed the predicted direction. Particularly strong results ($r > 0.50$) were obtained for power distance, individualism, individual autonomy, emancipative moral judgments, tightness, national levels of gender equality, and median income. Results for verbal confrontation were weaker and two of the correlations (for tightness and pro-violence attitudes) went weakly in the wrong direction. Thus, Hypothesis 5 received support but much more strongly for physical confrontation than for verbal confrontation, underscoring the need for making a distinction between these sanctions. Results for social ostracism showed the same pattern as for physical confrontation. However, results for gossip followed the exact opposite pattern. For example, the appropriateness rating of gossip was higher in countries that were higher on individualism, autonomy values, emancipative moral judgments, gender equality, and median income. This opposite pattern for gossip is consistent with our previous analysis of sanction-specificity of metanorms. (Correlations tend to keep the same signs when metanorms are estimated separately for non-cooperation and out-of-place behaviors, see Supplementary Table 6.)

When drawing conclusions about the origins of variation in metanorms, it is important to note that cultural, ecological, and economic variables are often strongly intercorrelated (Supplementary Table 7). Moreover, the strength of correlations between metanorms and emancipative moral judgments may in part be due to both constructs being based on appropriateness ratings of actions. Among the other variables, median income and the

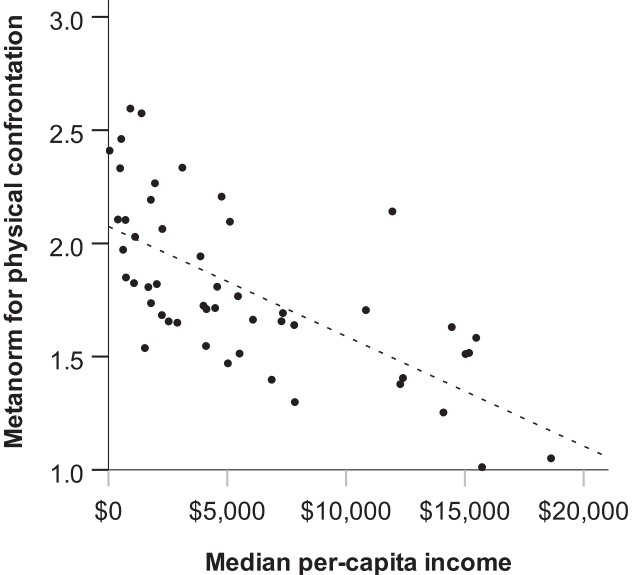

**Fig. 2 The negative association of median income with the appropriateness rating of physical confrontation across 50 countries.** Including regression line ($R^2 = 0.45$). Every dot represents a country. The x-axis represents median per-capita income according to Gallup[24]. The y-axis represents the metanorm for physical confrontation, that is, the mean appropriateness rating for scenarios where someone responds to a norm violation by physical confrontation.

national level of gender equality tended to show the strongest relation to metanorms overall. We use scatterplots to illustrate that in countries where median income was higher the perceived appropriateness of physical confrontation tended to be lower (Fig. 2), while the perceived appropriateness of gossip tended to be higher (Fig. 3).

## Discussion

Although norms about punishing norm violators may be critical for maintaining cooperation in human groups, there has been

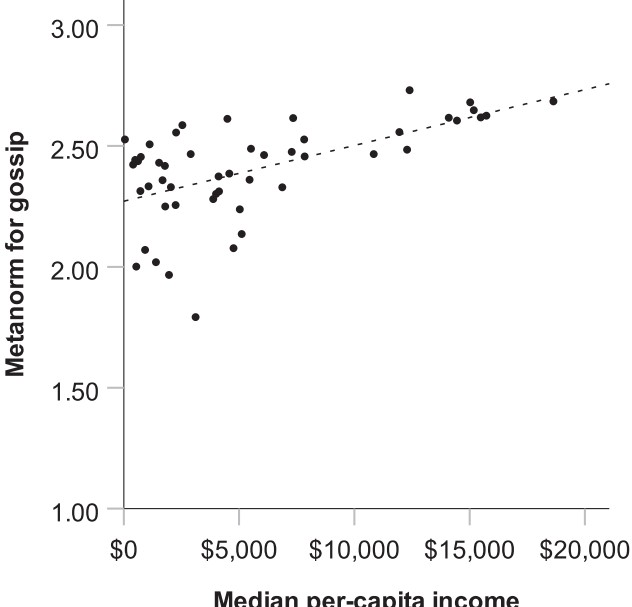

**Fig. 3 The positive association of median income with the appropriateness rating of gossip across 50 countries.** Including regression line ($R^2 = 0.33$). Every dot represents a country. The *x*-axis represents median per-capita income according to Gallup[24]. The *y*-axis represents the metanorm for gossip, that is, the mean appropriateness rating for scenarios where someone responds to a norm violation by gossip (adjusted for country differences in the appropriateness ratings of the norm violations).

little empirical research on which sanctions people in fact view as appropriate and how this may vary across different norm violations and across countries. The first key finding of this cross-cultural study of the perceived appropriateness of using informal sanctions was culture-universal: the participants consider it more appropriate to use gossip, social ostracism, and confrontation the more inappropriate the triggering behavior is perceived to be. This finding supports our assumption that these distinct responses are all universally used as expressions of disapproval and can therefore be conceived of as informal sanctions.

The next key finding was that metanorms for the different sanctions were consistent within countries and largely independent of the domain of the norm violation. Specifically, the same rules for what is an appropriate response to non-cooperation seem to apply to behavior that others simply find uncivil or out of place. This finding is consistent with a parsimonious psychology of informal sanctions that does not include any specific adaptations for the cooperative domain. It poses a challenge for theories of the evolution of cooperation, as it may not be sufficient to focus only on the cooperative domain when modeling the evolutionary dynamics of sanctions[29].

Our study also contributed to the longstanding debate on whether metanorms require punishment of norm violators. Theoretical work on altruistic punishment has often assumed that not punishing a norm violation is selfish and hence should be deemed inappropriate[30], and studies using economic experiments have found that those who pay a cost to punish others' selfish behavior are subsequently trusted more than non-punishers[31,32]. However, studies have also found that non-punishers are not viewed as more selfish[29,33] and do not elicit more disapproval[14,18–20]. Our finding that non-action was often viewed as the most appropriate response is consistent with this latter research, supporting the notion that metanorms often do not require bystanders to punish norm violators. Note that this conclusion applies only to relatively minor norm violations, as the perceived appropriateness of non-action was found to decrease for more serious infractions (Fig. 1).

When designing the study to include several forms of informal sanctions, it was an open question to us whether different forms would exhibit similar cross-cultural variation in appropriateness ratings. We speculated that there could instead be complementarity between preferences for confrontation and preferences for non-confrontational sanctions, such as social ostracism and gossip. Indeed, we did find a separation between societies condoning physical confrontation and societies condoning gossip—but, surprisingly, metanorms for gossip were negatively correlated with metanorms for social ostracism, which instead were positively correlated with metanorms for physical confrontation. It was also surprising that metanorms for physical and verbal confrontation were only weakly correlated. These results indicate that metanorms are sanction-specific. This interpretation was supported by the additional finding that metanorm measures for distinct sanctions correlated well with the reported levels of use of the same sanctions. Nonetheless, the observed pattern of consistencies and complementarities across different forms of informal sanctions remain intriguing puzzles that require further research. We offer some thoughts below.

By relating metanorms to other country variables, our study speaks to theories of how variation in metanorms may emerge. One theory is that variation in metanorms reflects variation in cultural values and norms; for instance, more individualistic values and loose norms may give individuals more leeway in violating norms without getting punished, while greater power distance may raise the acceptance of individuals asserting authority by punishing norm violators. Consistent with this theory, appropriateness ratings of physical confrontation and social ostracism were negatively correlated with individualism and looseness, and positively correlated with power distance. However, culture is not static. In a process thought to be driven by increasing economic prosperity, cultural values have been shifting quite rapidly in modern times, including increasing autonomy for individuals, more emancipative moral judgments, and less inequality between men and women[34]. Our study suggests that metanorms are similarly shifting. Although the shift itself cannot be observed in this cross-sectional study, we observed high positive correlations of metanorms with emancipative moral judgments, the national level of gender equality, and median income. An alternative theory is that both cultural values and metanorms respond to the local need for social coordination that may be caused by conditions of ecological threat, especially pathogen prevalence. Our data provided moderate support for the role of pathogen prevalence, but no support for perceived threat being related to metanorms.

Metanorms for gossip showed a unique pattern. In countries with higher median income, gossip tended to be more, not less, appropriate. Thus, if metanorms are indeed shifting as living standards rise in the population at large, gossip appears to become viewed as more appropriate as physical confrontation becomes viewed as less appropriate. The specific rise of the perceived appropriateness of gossip in countries with high living standards is one of the most intriguing findings of our study. What is it about gossip that makes its perceived appropriateness change in ways distinct from social ostracism, which is another non-confrontational sanction? One key difference is whether the response is directed to the norm violator or a third party. Specifically, confrontation and active avoidance concern responses that are related to how you interact with the norm violator; in contrast, gossiping concerns how you interact with another person. For this reason, people may think of both confrontation and social ostracism as "punishment" while viewing gossip in a

different way, even though they are all expressions of disapproval and even though gossip may be as effective in sustaining norms[35].

But why is gossip considered more appropriate, and "punishment" less appropriate, in societies that are more affluent and have more emancipative values? One possibility is that a decrease in the perceived appropriateness of "punishment" in these societies is compensated for by a complementary increase in the perceived appropriateness and use of gossip. Gossip allows one to examine whether other people share your evaluations and to prepare for alternative forms of communication, such as public messages that underscore a specific norm without singling out any individual[1]. Second, gossip may be viewed as more appropriate in more individualistic societies because of differences in social network structures; it may be viewed as less appropriate to talk about a norm violator to someone who is socially close to that person, which would typically be the case in collectivistic societies where social networks are more overlapping[36]. A third interpretation is that norms about gossip are less shaped by their role for norm enforcement than by their role in free information exchange, which arguably becomes more important as societal complexity increases[37].

Outside of lab experiments, we know of no data on the relative effectiveness of different forms of sanctions for achieving norm compliance. If some sanctions are more effective than others, the country variation we have observed may cause varying levels of compliance. This may be a particularly fruitful avenue of research in connection with the social norms emerging in response to the coronavirus crisis: Are violations of norms about social distancing, say, more common (or less common) in countries favoring gossip than in countries favoring confrontation?

Before closing, we should note some strengths and limitations of the present research. Although we used only 10 different norm violation scenarios, these covered a wide range of specific behaviors and contexts (e.g., singing in a library), supporting the exciting conclusion that metanorms apply across behaviors and contexts of the underlying norm violations, even though metanorms vary across countries. The scenarios were hypothetical, but results were validated against the actual use of informal sanctions reported by respondents. Finally, our sampling strategy had both strengths and limitations. By collecting data from both students and non-students, and across different cities, we established that these subsamples tended to have similar metanorms in the same country. However, it is possible that metanorms exhibit within-country variation along the urban–rural and socioeconomic dimensions, which we were unable to capture when focusing on urban locations with universities.

A major contribution of the present research is the finding that metanorms are not universal but are subject to systematic cross-societal variation. Note that a lack of consensus about the right way to deal with norm violators may contribute to conflict. Disagreement about social norms is a fact of social life. As the world becomes "smaller" and more interconnected, societies increasingly face the need to consider and negotiate what is the most appropriate response when one's own norm is violated. It also may make it more likely that one's own norm-violating behavior may elicit very different forms of sanctions. Both experiences underline not only the scientific importance of metanorms, but also how they may receive growing attention in a world that faces opportunities for cultural diversity.

## Methods

The study was preregistered at OSF (osf.io/qg6xy) at the start of data collection. The full survey and the data used in the present paper are openly available at OSF (osf.io/pm5kc/).

For comparability of samples, we set out to collect data from approximately 200 college students in a major city in each country, which was achieved in almost all countries. To assess the robustness of the country-level measures obtained from these samples, we complemented the main sampling strategy in two ways: (a) we collected additional data from non-student samples (or, in two cases, part-time students) in 31 countries; (b) we collected data from two or more student samples located in different cities of each of 10 countries. In total, we have data from 22,863 participants (students: $n = 18,091$; non-students: $n = 4772$), after excluding a few participants (1.5%) who reported an age under 18. Descriptions of the data collection sites and their sample characteristics are reported in Supplementary Table 1. Participants were recruited using a variety of methods, such as invitations via email, on social media, in class, face to face on campus, using public notices and flyers, and using survey organizations.

The survey was translated into 30 different languages, following the usual practice of independent translation and back-translation. The study was conducted anonymously online using Qualtrics, with a few exceptions. Part of the Estonian non-student sample and the Ghanaian student and non-student samples were collected using pen and paper at the university, with animations shown on a big screen.

All participants gave their informed consent and we complied with all relevant ethical regulations. Approval of the study protocol was obtained from ethics committees and institutional review boards where required, including Queen's University (Canada), York University (Canada), Bogotá (Colombia), Institute of Psychology at the Czech Academy of Sciences (Czech Republic), Universidad San Francisco de Quito (Ecuador), United Psychological Research Committee (Hungary), Monk Prayogshala (India), the Trinity College Dublin School of Social Sciences and Philosophy (Ireland), Kwansei Gakuin University (Japan), Aoyama Gakuin University (Japan), United States International University – Africa (Kenya), Sunway University (Malaysia), University of Amsterdam (Netherlands), Komisja ds. Etyki Badań Naukowych Wydziału Psychologii Uniwersytetu SWPS (Poland), Instituto de Ciências Sociais (Portugal), Doha Institute for Graduate Studies (Qatar), Singapore Management University (Singapore), Sungkyunkwan University (South Korea), Universidad de Navarra (Spain), Post Graduate Institute of Medicine (Sri Lanka), Chulalongkorn University (Thailand), American University of Sharjah (United Arab Emirates), University of Kent (United Kingdom), Brunel College of Health and Life Sciences (United Kingdom), University of South Carolina (United States), and New York University (United States).

**Scenarios**. Scenarios were selected to cover potentially norm-violating behavior in three domains: cooperation, out-of-place everyday behavior, and meta-violations (i.e., potentially norm-violating use of an informal sanction).

The cooperative domain was covered by an animation of an agent depleting a common resource, referred to as scenario A. This scenario was drawn from prior research on metanorms[14].

Out-of-place everyday behavior was covered by four scenarios describing someone (B) listening to music on headphones at a funeral, (C) sleeping in a restaurant, (D) singing in a library, or (E) reading a newspaper at the movies. These combinations of behaviors and contexts were found to be widely viewed as inappropriate in a prior cross-cultural study of norms[16].

Meta-violations included two instances of physical confrontation: (F) an animation of an agent physically confronting someone who depleted a common resource in scenario A, and (G) a verbal scenario with a man being physically aggressive against someone who insulted his mother. We use scenarios F and G to calculate metanorm measures for physical confrontation.

The remaining three meta-violation scenarios described someone who reacted to a person who was rude in a public place in one of three ways: (H) by reprimanding this person, (I) by speaking negatively about this person, or (J) by staying away from this person.

**Missing values**. Missing values were handled by imputation, using the EM method in SPSS.

**Standardization**. To control for response sets with respect to the appropriateness response scale, the preregistered plan specified standardization of the participants' mean response across all items, referring to the 50 items of the metanorm instrument, which all used the same response scale from extremely inappropriate to extremely appropriate. Notably, in addition to the metanorm instrument, the survey included various other items that used different response scales to measure how often something happens or how strongly the respondent agrees with a statement, etc. All 50 appropriateness ratings of a participant were adjusted by a constant equal to the grand mean of all appropriateness ratings in the entire sample minus the mean of all appropriateness ratings by that participant. Thus, ratings were raised for participants who had tended to use lower ratings than the average participant, while ratings were lowered for participants who had tended to use higher ratings than the average participant. The standardized ratings have the property that the mean rating across the 50 appropriateness items is identical for every participant (and identical to the grand mean of the original ratings across the entire sample).

When interpreting results based on standardized ratings, we account for the fact that standardization leads to some artificial negative effects on correlations between different appropriateness items (i.e., items that are in fact uncorrelated will, after standardization, tend to become slightly negatively correlated). Below we also

consider an additional standardizing method that was not preregistered: standardizing metanorm measures for sanctions by subtracting the metanorm measure for non-action.

**Calculation of metanorm measures**. As specified in the preregistered analysis plan, metanorm measures were obtained by adjusting county mean ratings for a given response (verbal confrontation, social ostracism, gossip, or non-action) by controlling for individual appropriateness ratings of the underlying norm violations. The technical specification is as follows.

Let $N_{s,c,i}$ denote the appropriateness rating of the norm violation in scenario $s$ given by individual $i$ in country $c$ (centered on the global mean). Let $N_{s,c}$ denote the average value of $N_{s,c,i}$ over all respondents from country $c$. Let $R_{s,c,i}$ denote the appropriateness of the given response in scenario $s$ as rated by individual $i$ in country $c$. Then the metanorm measure in country $c$, denoted by $R_c$, is calculated by estimating the multi-level model

$$R_{s,c,i} = R_c + b_1 N_{s,c} + b_2 N_{s,c,i} + e_{c,i} + e_{s,c,i}, \qquad (1)$$

where the terms $b_1 N_{s,c} + b_2 N_{s,c,i}$ adjust for the appropriateness rating of the norm violation at country- and individual level, $e_{c,i}$ is a random effect at the individual level, and $e_{s,c,i}$ is the residual error term.

Scenarios A–E were used in the main estimation. However, other sets of scenarios may be used instead. Robustness checks reported in the main text included basing metanorm measures on the set of all ten scenarios (A–J) as well as only on scenario A. Note that when a single scenario is used, the country-level term $b_1 N_{s,c}$ becomes redundant and the multi-level model reduces to a simple linear regression.

**Culture measures**. The survey included the following culture measures.

*Hofstede scales*. Four-item scales for individualism, power distance, and indulgence (12 items in total) from the Hofstede VSM 2013 questionnaire. Country-mean responses showed all three scales had poor internal consistency, all $\alpha < 0.30$, so they are not used.

*Use of informal sanctions*. Single items on participants' own use of confrontations, gossip, and avoidance (e.g., "How often does someone confront you for doing something inappropriate?" and "How often do you confront someone who does something inappropriate?"), and on participants' perceptions of others' use of these sanctions against themselves (e.g., "How often does someone confront you for doing something inappropriate?"), on a five-point scale from "never" (1) to "always" (5). We use the country-mean responses.

*Individual autonomy*. We use a measure of cultural values on individual autonomy adopted from the World Values Survey (WVS). Participants are asked to select up to five important qualities for children to learn at home, from a list of 10 qualities. Among the potential alternatives are independence, determination/perseverance, religious faith, and obedience. As in the WVS, the autonomy measure (ranging from −2 to +2) was calculated by the formula *Autonomy = Independence + Determination – Faith – Obedience*, where qualities are coded 1 if selected, 0 otherwise. At the country level this measure had adequate internal consistency ($\alpha = 0.75$).

*Emancipative moral judgments*. We use a four-item scale adopted from the WVS, asking how justified it is with homosexuality, divorce, abortion, and suicide, on a scale from never justified (0) to always justified (10). Country-level internal consistency was very good ($\alpha = 0.92$).

*Pro-violence attitudes*. We similarly use two items adopted from the WVS, asking how justified it is for a man to beat his wife and to use violence against other people ($\alpha = 0.78$).

*Tightness*. We use Gelfand's 6-item tightness scale[16], with items like "There are many social norms that people are supposed to abide by in this country." In the original study, responses were standardized by subtracting participants' mean response to all items in the survey, which was strongly dominated by items on the appropriateness of various behaviors in various contexts. Following this procedure, we adjusted the responses to the tightness items in our survey by subtracting participants' mean response to all appropriateness items. Country-level internal consistency was good ($\alpha = 0.80$).

*Perceived threat*. To measure perceived threat we included a question original to this study: "Which of the following do you think are threats to your society (tick any that apply): conflict within the country, conflict with other countries, immigration, over-population, food deprivation, lack of safe water, poor quality of air, natural disasters, diseases?". A tick for a given threat was coded as 1, no tick as zero. Country-means had good internal consistency ($\alpha = 0.89$).

**Attention and comprehension**. Measures of attention and comprehension were included at the end of the survey. The attention test asked the participant to tick the fourth box out of five. The comprehension test asked the participant how easy or difficult it had been to understand the questions in the survey, on a five-point scale from very difficult to very easy. In the robustness check reported in the main text we excluded participants who had not answered one or both of these questions (21.1%), or ticked the wrong box in the attention test (an additional 1.0%), or answered that it was very difficult to understand the survey (an additional 0.4%).

**Changes to the preregistered analyses**. The present paper presents the preregistered analyses with the following three changes.

*Exclusions*. No exclusions were planned, but as the study was meant to target adults, we decided to exclude respondents who stated an age below 18 years.

*Measures of indulgence, power distance, and individualism*. Because these scales turned out to lack adequate reliability, we instead decided to use the official Hofstede Insights country scores (obtained from www.hofstede-insights.com/product/compare-countries/) for these cultural dimensions. Although still widely used in research, a drawback is that these country scores typically build on data collected long ago, especially for power distance and individualism, and may not reflect recent cultural changes[38].

*The use of informal sanctions*. To measure the use of informal sanctions, we decided to focus on participants' reports of own use of sanctions and disregard their perceptions of how often they were sanctioned by others, as it is unlikely that people have accurate perceptions of how much others avoid them or gossip about them.

**Unregistered analyses**. The main text describes some elements that were not preregistered: inclusion of pathogen prevalence, median income, and the national level of gender equality as correlates in Hypothesis 5; calculation of metanorms for physical confrontation; robustness of metanorms across different cities and across student and non-students; internal consistency of a metanorm across scenarios; robustness of correlations with other variables whether metanorms are estimated in the domain of non-cooperation or the domain of out-of-place behaviors. As an additional unregistered analysis, metanorms for informal sanctions were standardized by the metanorm for non-action. Specifically, subtraction of the metanorm for non-action from the metanorms for sanctions was carried out to yield a measure of how appropriate the sanction is perceived to be relative to doing nothing at all. This method has the drawback that ratings for non-action exhibit meaningful country variation (as seen in Table 2), which will be incorporated in the measures for every sanction, thereby making them artificially more closely intercorrelated. Nonetheless, the pattern of results for how metanorms vary across cultures remains qualitatively the same (see Supplementary Table 8).

**Reporting summary**. Further information on research design is available in the Nature Research Reporting Summary linked to this article.

## Data availability
All data and materials are available at OSF (https://osf.io/pm5kc/), including the raw data underlying Figs. 1–3 and SPSS syntax for analyses. A reporting summary for this Article is available as a Supplementary Information file. Source data are provided with this paper.

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

## Acknowledgements
This research was funded by the Swedish Foundation for Humanities and Social Sciences (Riksbankens Jubileumsfond) [P17-0030:1]. The contributions of S.G. and M.H. were supported by the Czech Science Foundation [20-01214S] and the Institute of Psychology, Czech Academy of Sciences [RVO: 68081740].

## Author contributions
K.E. designed the study, analyzed the data, and wrote the manuscript. P.S., M.G., P.A.M.V.L., J.W. and B.S. provided critical input on the study and the manuscript. J.W., J.A., C.S.A., Al.A., P.A.A., G.A., A.d.A., G.A., Z.A., F.B., D.B., D.B.-B., B.B., A.B., E.B., M.B., S.B., P.B., F.B.Z., I.B., B.T.T.H., J.-C.C., Đ.Č., H.-S.C., C.C.C.-I., R.C.-L., M.d.B., P.d.Z., A.D., N.D., A.E., J.B.E., H.E., X.F., S.F., O.A.F.-G., M.F., R.B.G., C.M.H.D.G., And.G., S.G., Ani. G., V.G., K.G., P.H., A.H., T.H., M.H., D.I., H.I., H.K., K.K., Nar.K., Nat.K., Ni.K., T.K., M.K., L.T.L., L.M.L., Y.L., N.P.L., Z.L., K.L., A.T.M., B.M., H.M., I.M., S.M., L.M., P.N., O. N., R.N., N.G.O., I.E.O., S.Ö., P.P., L.R.P.-F., Mi.P., Mp.P., A.-M.P.-B., Ma.P., J.R., C.R., R.B.R., S.R., P.P.R., I.S., A.S.M., S.S., H.S., B.S., E.S., K.T., Has.T., M.L.M.T., N.T., Hab.T., G.A.T., Y.T., R.W., S.W., R.Z., Q.-p.Z. and L.Z.-K. collected data. All authors reviewed the manuscript.

## Funding

## Competing interests
The authors declare no competing interests.

## Additional information

Kimmo Eriksson [1,2✉], Pontus Strimling[3], Michele Gelfand[4], Junhui Wu[5], Jered Abernathy[6], Charity S. Akotia[7], Alisher Aldashev[8], Per A. Andersson[1,9], Giulia Andrighetto[2,3,10], Adote Anum[7], Gizem Arikan [11], Zeynep Aycan[12], Fatemeh Bagherian[13], Davide Barrera [14], Dana Basnight-Brown [15], Birzhan Batkeyev[16],

Anabel Belaus [17,18], Elizaveta Berezina [19], Marie Björnstjerna[3], Sheyla Blumen [20], Paweł Boski[21], Fouad Bou Zeineddine [22], Inna Bovina [23], Bui Thi Thu Huyen[24], Juan-Camilo Cardenas [25], Đorđe Čekrlija[26], Hoon-Seok Choi[27], Carlos C. Contreras-Ibáñez [28], Rui Costa-Lopes[29], Mícheál de Barra[30], Piyanjali de Zoysa[31], Angela Dorrough[32], Nikolay Dvoryanchikov [23], Anja Eller[33], Jan B. Engelmann [34], Hyun Euh [35], Xia Fang [36], Susann Fiedler[37], Olivia A. Foster-Gimbel [38], Márta Fülöp[39,40], Ragna B. Gardarsdottir [41], C. M. Hew D. Gill [19], Andreas Glöckner[32,37], Sylvie Graf [42], Ani Grigoryan [43], Vladimir Gritskov [44], Katarzyna Growiec [21], Peter Halama [45], Andree Hartanto[46], Tim Hopthrow [47], Martina Hřebíčková [42], Dzintra Iliško[48], Hirotaka Imada [47], Hansika Kapoor [49], Kerry Kawakami[36], Narine Khachatryan [43], Natalia Kharchenko[50], Ninetta Khoury[51], Toko Kiyonari[52], Michal Kohút [53], Lê Thuỳ Linh[54], Lisa M. Leslie[38], Yang Li [55,56], Norman P. Li[46], Zhuo Li [57], Kadi Liik[58], Angela T. Maitner [59], Bernardo Manhique[60], Harry Manley [61], Imed Medhioub [62], Sari Mentser[63], Linda Mohammed[64], Pegah Nejat [13], Orlando Nipassa[60], Ravit Nussinson [63,65], Nneoma G. Onyedire [66], Ike E. Onyishi[66], Seniha Özden[12], Penny Panagiotopoulou[67], Lorena R. Perez-Floriano[68], Minna S. Persson[3], Mpho Pheko[69], Anna-Maija Pirttilä-Backman[70], Marianna Pogosyan[71], Jana Raver[72], Cecilia Reyna[17], Ricardo Borges Rodrigues [73], Sara Romanò[74], Pedro P. Romero [75], Inari Sakki [76], Alvaro San Martin[77], Sara Sherbaji [59], Hiroshi Shimizu[78], Brent Simpson[6], Erna Szabo[79], Kosuke Takemura[80], Hassan Tieffi[81], Maria Luisa Mendes Teixeira[82], Napoj Thanomkul[61], Habib Tiliouine [83], Giovanni A. Travaglino [47,84], Yannis Tsirbas[85], Richard Wan[79], Sita Widodo[86], Rizqy Zein [86], Qing-peng Zhang[87], Lina Zirganou-Kazolea[85] & Paul A. M. Van Lange [88]

[1]Center for Cultural Evolution, Stockholm University, Stockholm, Sweden. [2]Mälardalen University, Västerås, Sweden. [3]Institute for Futures Studies, Box 591 Stockholm, Sweden. [4]Department of Psychology, University of Maryland, College Park, MD, USA. [5]CAS Key Laboratory of Behavioral Sciences, Chinese Academy of Sciences, Chaoyang District, Beijing, China. [6]Department of Sociology, University of South Carolina, Columbia, SC, USA. [7]Department of Psychology, University of Ghana, P.O. Box LG 84 Legon Accra, Ghana. [8]New School of Economics, Satbayev University, Almaty, Kazakhstan. [9]Department of Behavioral Sciences and Learning, Linköping University, Linköping, Sweden. [10]Institute of Cognitive Sciences and Technologies, National Research Council of Italy, Rome, Italy. [11]Department of Political Science, Trinity College Dublin, Dublin 2, Ireland. [12]Koç University, Rumelifeneri, Sarıyer Rumelifeneri Yolu, Istanbul, Turkey. [13]Department of Psychology and Education, Shahid Beheshti University, Tehran, Iran. [14]University of Turin and Collegio Carlo Alberto, Turin, Italy. [15]United States International University – Africa, Box 14634 00800 Nairobi, Kenya. [16]International School of Economics, Kazakh-British Technical University, Almaty, Kazakhstan. [17]Instituto de Investigaciones Psicológicas (IIPsi), Consejo Nacional de Investigaciones Científicas y Técnicas (CONICET), CABA, República Argentina. [18]Universidad Nacional de Córdoba (UNC). Facultad de Psicología (UNC), Ciudad Universitaria, Bv. de la Reforma esquina, Enfermera Gordillo s/n, Córdoba, Argentina. [19]Sunway University, No. 5, Jalan Universiti, Bandar Sunway, Selangor, Darul Ehsan, Malaysia. [20]Departamento de Psicología, Pontificia Universidad Católica del Perú, San Miguel, Lima, Peru. [21]SWPS University of Social Sciences and Humanities, Warsaw, Chodakowska, Poland. [22]Department of Psychology, University of Innsbruck, Innsbruck, Austria. [23]Moscow State University of Psychology and Education, Moscow, Russia. [24]Hanoi National University of Education, Cau Giay District, Hanoi, Vietnam. [25]Universidad de los Andes, Colombia, Bogota, Colombia. [26]Faculty of philosophy, University of Banja Luka, Banja Luka, Bosnia and Herzegovina. [27]Department of Psychology, Sungkyunkwan University, Seoul, Republic of Korea. [28]Departamento de Sociología, Universidad Autónoma Metropolitana - Unidad Iztapalapa, Ciudad de México, Mexico. [29]Instituto de Ciências Sociais, Universidade de Lisboa, Lisboa, Portugal. [30]Center for Culture and Evolution, Brunel University London, Uxbridge, UK. [31]Faculty of Medicine, University of Colombo, Colombo 8, Sri Lanka. [32]Department of Psychology, University of Cologne, Cologne, Germany. [33]Facultad de Psicología, Universidad Nacional Autónoma de México. Av. Universidad 3004, Ciudad Universitaria, Ciudad de México, Mexico. [34]Center for Research in Experimental Economics and Political Decision Making (CREED), Amsterdam School of Economics, University of Amsterdam, P.O. Box 15867 Amsterdam, NJ, The Netherlands. [35]Department of Psychology, University of Minnesota, Minneapolis, MN 55455, USA. [36]Department of Psychology, York University, Toronto, ON, Canada. [37]Max Planck Institute for Research on Collective Goods, Bonn, Germany. [38]New York University, Stern School of Business, New York, NY 10012, USA. [39]Institute for Cognitive Neuroscience and Psychology, Research Centre of Natural Sciences, Budapest, Hungary. [40]Eötvös Loránd University, Faculty of Psychology and Education, Budapest, Hungary. [41]Department of Psychology, University of Iceland, Reykjavík, Iceland. [42]Institute of Psychology, Czech Academy of Sciences, Brno, Czech Republic. [43]Department of Personality Psychology, Yerevan State University, Yerevan, Armenia. [44]Saint Petersburg State University, St Petersburg, Russia. [45]Center for Social and Psychological Sciences, Slovak Academy of Sciences, Bratislava, Slovakia. [46]School of Social Sciences, Singapore Management University, Singapore, Singapore. [47]School of Psychology, University of Kent, Canterbury, UK. [48]Daugavpils University, Daugavpils, Latvia. [49]Department of Psychology, Mumbai, Maharashtra, India. [50]Kyiv International Institute of Sociology, Kyiv, Ukraine. [51]Future Minds Gifted Centre, Lima, Peru. [52]Aoyama Gakuin University, Sagamihara-city, Kanagawa, Japan. [53]Faculty of Philosophy and Arts, University of Trnava, Trnava, Slovakia. [54]National Economics University, Hai Ba Trung, Dong Tam District, Hanoi, Vietnam. [55]Nagoya University, Furo-cho, Chikusa-ku, Nagoya, Aichi, Japan. [56]Melbourne School of Psychological Science, University of Melbourne, Parkville, VIC, Australia. [57]Department of Psychology, University of Western Ontario, London, ON, Canada. [58]School of Natural Sciences and Health, Tallinn University, Tallinn, Estonia. [59]Department of International Studies, American University of Sharjah, PO Box 26666 Sharjah, United Arab Emirates. [60]Eduardo Mondlane University, Faculty of Arts and Social Sciences, Department of Sociology, Maputo, Mozambique. [61]Faculty of Psychology, Chulalongkorn University, Bangkok, Thailand. [62]Department of Finance and Investment, Al Imam Mohammad Ibn Saud Islamic University (IMSIU), P.O. Box 5701 Riyadh,

Saudi Arabia. [63]The Open University of Israel, Raanana, Israel. [64]Institute of Criminology and Public Safety, Valsayn Campus, Graver Road, Valsayn, University of Trinidad and Tobago, Arima, Trinidad and Tobago. [65]University of Haifa, Haifa, Israel. [66]Department of Psychology, University of Nigeria Nsukka, Nsukka, Nigeria. [67]Department of Education and Social Work, University of Patras, Rion, Patras, Greece. [68]Universidad Diego Portales, Santiago, Chile. [69]Department of Psychology, University of Botswana, Private Bag UB 00705, Gaborone, Botswana. [70]University of Helsinki, Faculty of Social Sciences, Social Psychology, PO Box 54 (Unioninkatu 37) Helsinki, Finland. [71]Politics, Psychology, Law and Economics (PPLE), University of Amsterdam, PO Box 15575 Amsterdam, The Netherlands. [72]Queen's University, Goodes Hall, Queen's University, Kingston, ON, Canada. [73]Instituto Universitário de Lisboa ISCTE-IUL, CIS, Lisbon, Portugal. [74]Department of Culture, Politics and Society, University of Turin, Turin, Italy. [75]Experimental and Computational Economics Lab (ECEL), School of Economics, Universidad San Francisco de Quito, Diego de Robles y Pampite, Quito, Ecuador. [76]University of Eastern Finland, Department of Social Sciences, P.O. Box 162770211 Kuopio, Finland. [77]IESE Business School, Madrid, Spain. [78]Kwansei Gakuin University, Nishinomiya, Hyogo, Japan. [79]Department of International Management, Johannes Kepler University, Linz, Austria. [80]Faculty of Economics, Shiga University, Hikone, Shiga, Japan. [81]Université Félix Houphouët-Boigny Cocody-Abidjan, Centre Ivoirien d'Etude et de Recherche en Psychologie Appliquée (CIERPA), Abidjan, Côte d'Ivoire. [82]Mackenzie Presbyterian University, Business Administration Postgraduate Program, São Paulo, Brazil. [83]Labo-PECS, Faculty of Social Sciences, Université d'Oran 2, Oran, Algeria. [84]School of Humanities and Social Science, The Chinese University of Hong Kong, Shenzhen, Longgang District, Shenzhen, P. R. China. [85]University of Athens, Department of Political Science and Public Administration, Athens, Greece. [86]Department of Personality and Social Psychology, Universitas Airlangga, Surabaya, Indonesia. [87]Guangzhou University, Guangzhou Higher Education Mega Center, Guangzhou, P.R. China. [88]VU Amsterdam, Department of Experimental and Applied Psychology, Institute for Brain and Behavior Amsterdam (IBBA), Amsterdam, Amsterdam, The Netherlands. ✉email: kimmoe@gmail.com

