## [Peer Review File · Nature Communications]

REVIEWER COMMENTS

Reviewer #1 (Remarks to the Author):

This cross-site cross-cultural study adds to our understanding of how societies maintain cooperation using norms about how to enforce norms. The work shows some interesting insights into how commonalities of normative influence are maintained across different groups as explained by cross-cultural dimensions of variation.

My major concern about this paper has to do with the way that the analyses are described in the main text. I find it hard to follow the explanation here as to how and why country means were used instead of retaining the full information of the data from individual respondents. In the method section, some but not all of the means used in the analyses are explained as derived from multilevel models and (might?) be empirical Bayes means. I see references to the preregistration, but I think it would do to repeat some of the key methods here at least in the supplement as the preregistration documentation could become hard to follow up as this publication gets older.

As a similar but more minor point, the axes in figure 1 are unclear and it would be much easier to read this plot with labels on the axes.

I also find the method detail on when and how 40-year-old aggregate data on cultural dimensions were used vs. the new measures of these dimensions from the data collected for this study. It is unclear to me which countries had new dimensional data collected, which were reliant on the old data, and which were imputed. Perhaps a table added to the supplement would help.

I also think it would also be informative to add some kind of figure like a map showing the relative proportion of the sample collected from different regions of the world. There clearly are some non-WEIRD populations included, though all appear to be from either student and/or urban centres. Having some indication of how widely the cross-cultural sample expands beyond the usual Western world would be helpful.

Reviewer #2 (Remarks to the Author):

This is a very interested and well-conducted paper. The authors examine how people respond to rule-breakers across societies, and they show how different types of reactions are predicted by country-level factors such as median income, individualistic values, and gender equality. This is a topic of great theoretical importance: in addition to being a cross-cultural comparison of norms and the socio-ecological causes thereof (which is novel unto itself), it also speaks towards the literature on punishment and what maintains certain norms in different societies. This is an ambitious project, with data from an impressive 57 countries, and at least 88 co-authors. Altogether, this is a great piece of work and I think it is highly appropriate for Nature Communications and will be well-cited. I have some recommendations for minor revisions, but the authors should be able to fix these upon revision.

My two biggest comments are as follows:

First, many countries rate the most appropriate action as "non-action". This warrants discussion. What does this say about whether there is indeed a meta-norm? The results are interesting either way, but the authors might need to tone down whether or not the meta-norm does or does not support the norm.

Second, I worry that the authors are extrapolating too much from the proportion of variance explained by country. This is just a small point in main text (an analysis of a secondary or tertiary point), but it is relevant to some larger debates, and it may be extrapolating farther than it should. First, the

analysis across cities only includes student samples. We should expect some similarity in student samples within a country, not only because there may be a common "student culture", but also because there is often mixing of students from different regions, i.e., students at a given university may come from many regions, which decreases the variance among "cities" within a country. Second, the comparison of student & non-student samples is all from people residing in cities. In many countries, the culture is different within cities than it is in rural areas. For example, in the USA, the Republican vs. Democrat divide is less of a state-by-state difference than it is a rural vs. urban difference. A similar pattern holds in Canada, with Conservatives dominating outside of urban centres. Similarly, analyses of Ultimatum Games show that while participants in industrialized countries behave similarly by making & demanding fair offers, participants in small-scale societies often behave differently by making & tolerating less fair offers (see Joe Henrich's work). Other work has shown very different behaviours in different parts of the same city, whereby affluent areas are more cooperative (for example) than less affluent areas (see Daniel Nettle's work in Newcastle UK; I believe David Sloan Wilson may have similar patterns in Binghamton NY). Thus, depending on how the researchers in each country sampled the non-student populations, it might have resulted in more homogeneity if the non-student samples were collected from more affluent parts of town surrounding the universities. This will reduce the variance within countries, making the country-level variance appear higher than it actually is. If the authors wish to make strong claims about the proportion of variance attributable to country vs. city or student-non-student, then we need more information on how each sample recruited non-student participants. Otherwise, the authors must present significant cautions around the interpretation of such results, for example a summary of the above points. These cautions must be in the main text, though it could just be a short summary with more elaboration in the Supplement next to Table S5.

I should note that it's good that the authors only compared the 10 countries that had >1 city, otherwise the variance accounted by country would have been inflated by the fact that only 10/57 countries had >1 city. Same with only comparing the 31 countries with student + non-student samples.

Third, the authors should clarify if they get the same results if standardizing with the rating for non-action instead of average of all items. I believe this is done somewhere, and if so, it should come out clearer in the writing, preferably early (i.e., when describing standardizing based on averages).

The following are all very minor comments:

- Figure 1 should have some title on y axis
- Specify boxes and whiskers in figure 1
- When discussing the lack of global consensus, the authors should specify that some countries had no sanction as most appropriate action
- Line 502 (Methods) says the controlled metanorm estimates are in Table S4, but it's actually S3
- Table S7 is potentially very useful for other researchers. To facilitate meta-analyses, the authors should include 95% CI's or something similar to give an estimate of the uncertainty. (This should probably be included anyway, but will be especially useful for future meta-analyses.)
- On lines 284-286, the authors claim that non-punishers are not viewed harshly in other literature. While this is strictly true that non-punishers are not punished, multiple studies going back a couple decades show that non-punishers are trusted less than are punishers. These include:
 - Barclay, P. (2006). Reputational benefits for altruistic punishment. *Evolution and Human Behavior*, 27, 325-344.
 - Nelissen, R. (2008). The price you pay: cost-dependent reputation effects of altruistic punishment. *Evolution & Human Behavior*, 29(4), 242-248.
 - Raihani, N.J., & Bshary, R. (2015). Third-party punishers are rewarded, but third-party helpers even more so. *Evolution*, 69, 993-1003.
 - Jordan, J.J., Hoffman, M., Bloom, P., & Rand, D.G. (2016). Third-party punishment as a costly signal of trustworthiness. *Nature*, 530, 473-476.

See also: Raihani, N.J., & Bshary, R. (2014). The reputation of punishers. *Trends in Ecology & Evolution*, 30(2), 98-103.
This work should be mentioned.

I think that the authors can address these comments upon review. If they can do so, then I think that this paper will make a strong addition to the literature. I am happy to elaborate on any of my comments if the authors wish.

Signed,
Pat Barclay, University of Guelph, barclayp@uoguelph.ca

Reviewer #3 (Remarks to the Author):

This paper is an impressive undertaking. The authors ran a preregistered study on the appropriateness of different responses to norm violations in 57 countries, translating the survey into 30 languages and, when possible, running it with both student and non-student samples. The authors find that people in different societies vary in how much they endorse different responses to norm violations and that these differences correlate, to some extent, with ecological, institutional, and cultural variables.

The data are important and relevant to ongoing debates in the literature on norm enforcement, cooperation, and cross-cultural variation. But, as currently written, the paper suffers from several shortcomings:

1. The biggest limitation of the manuscript is that it isn't clear what the theoretical stakes are. The authors write that previous work has shown that norms governing punishment vary with individualism and tightness and that these are linked to other institutional and ecological factors. They go on to show that physical sanctions exhibit these patterns, verbal sanctions less so. That's fine, of course, but it's not clear why these results matter. How should the reader interpret the fact that tightness, etc. seem to affect physical sanctioning but not verbal sanctioning?

I recommend that the authors beef up the theoretical presentation in the introduction. To say that they make predictions based on results in previous papers is fine, but it would be much more compelling and important if they grounded this in larger theory. Perhaps the theory is that, for whatever reason (cultural group selection? Adaptive psychology?), under conditions of greater social and ecological threat, societies enforce norms more tightly (to promote coordination?). Perhaps this theoretical grounding makes a suite of predictions that the authors can precisely specify (such as in a table). Perhaps there are reasons to expect that gossip, physical sanctioning, ostracism, and verbal sanctioning should all respond similarly to tightness and social ecological threat; perhaps there are reasons not to expect similarities. Regardless, the paper would benefit from being clearer on which theory is being tested and what the predictions of that theory are. This would also mean editing the discussion, pointing out which results were predicted, which results violated predictions, and what that might mean. (In that vein, it now reads like the gossip results were surprising, but it's not clear whether the "hypotheses" made predictions about gossip, or whether those are more exploratory.)

2. Related to the last point, it wasn't clear to what extent the predicted relationship between tightness/threat, on the one hand, and appropriateness of norm enforcement, on the other, was supported. Tightness was positively correlated with the acceptance of physical confrontation yet seemed uncorrelated with meta-norms about verbal sanctioning; the same was true for gender equality. Perceived threat seemed uncorrelated with metanorms for both physical sanctioning and verbal sanctioning. This seems either inconvenient for the authors or theoretically interesting (or both). Regardless, it should be discussed.

EDIT: I see that the preregistration presents clear hypotheses with theoretically rooted justifications

and precise predictions. These should be included in the paper. If the authors choose a set of hypotheses or predictions aside from those in the pre-registration, or if they choose not to present some of them, then that divergence should be justified.

3. It seemed odd that gossip was presented as a sanction. People don't seem to gossip as a way of incentivizing good behavior; they seem to gossip to share and acquire information. The fact that gossip incentivizes norm compliance seems incidental. (Relatedly, the idea expressed in lines 250-252 that a correlation between the appropriateness of some behavior X and the inappropriateness of a norm violation shows that X is a sanction seemed odd, given that other behaviors (such as a desire to acquire information) seem like they would show a similar pattern.)

4. Physical sanction is often excluded from comparisons among the other reactions to norm violations (e.g., in Figure 1). Why is this? If it's because the physical sanctioning data could not be compared to the other data because of how they were collected, that might be worth saying early in the paper (although it isn't clear why they were comparable for Figure 2 but not comparable for Figure 1). Without addressing early in the paper why physical sanctioning data are only sometimes presented, readers may suspect that the authors are intentionally excluding data.

Smaller comments:

1. Regarding this statement, "Prior cross-cultural work has not distinguished between forms of sanctions", I was surprised to read that. If it's true, that adds to the importance of the paper. I was quickly reminded of this pre-print by Moya et al., <https://drive.google.com/file/d/11tIKFkScdggCvgsu-Qr7XgGKur-SA7jW/view>, although the authors may not count it for some reason (e.g., it's still in the review process). Regardless, some readers may find this sentence surprising, so it seems worth explaining it in greater depth. The authors cited papers by Brauer & Chaurand, Eriksson et al., and Gelfand et al., but it would help to be clear on whether other cross-cultural projects on norms and norm enforcement (e.g., by House et al., Henrich et al., Barrett et al., Schulz) also fail to distinguish between forms of sanctions.

2. This sentence seemed unjustified: "Finally, by collecting data from both students and nonstudents, and across different locations in the same country, we established that metanorms look pretty much the same in different groups within a society, even though they differ across societies."

It's important and valuable that the authors collected non-student samples, but this was done in a subset of the countries and the non-student samples looked like they often (always?) came from the same urban areas as the student samples (in two cases, it sounded like, they were collected at universities). Surveying English-speaking students and non-students in Mumbai, for instance, doesn't seem like it shows that metanorms look pretty much the same across India.

Reviewer #4 (Remarks to the Author):

This manuscript describes an impressive study of meta-norms about peer punishment across a diverse set of 57 societies, with a large number of participants (N = 22,863). It examines how the appropriateness of norm violations and distinct responses to norm violations (confrontation, gossip, and ostracism) varies across cultures, and how cultural (e.g., individualism, tightness) and environmental (e.g., pathogen stress, threat estimates) factors relate to norms about more vs. less confrontational punishment of rule-breakers.

The authors provide compelling evidence for cultural universals and cultural variation in meta-norms. They find that, across societies, people view the use of verbal confrontation, gossip, and ostracism as more acceptable, the more inappropriate a norm violation. At the same time, they observe substantial

cross-cultural variation in the types of punishment that are deemed most appropriate: while some societies favor confrontational strategies, others favor indirect strategies of gossip or ostracism. Importantly, cultural and environmental factors are used to explain this variation. As a case in point, median per-capita income and gender gap indicators differentially relate to meta-norms about confrontation versus gossip. In countries with higher median income and gender equality, the acceptability of confrontation is lower while the acceptability of gossip is higher.

In my view, these results offer novel insights, pertinent to our understanding of the evolution of cooperation, the functions of punishment strategies, and cultural universality and variability. Findings underscore the importance of studying norms about distinct types of punishment (moving beyond a current focus only on costly, direct punishment). They provide support for the exciting possibility that different societies find different solutions – by developing norms that favor distinct punishment strategies – to promote cooperation and deter rule-breakers. And they move one step further to examine which societal-level factors relate to meta-norms about distinct forms of punishment. These findings have the potential to be highly influential among researchers in different fields—evolutionary biology, cultural evolution, psychology, and behavioral economics.

Going into the nuts and bolts of the paper, there are multiple aspects to appreciate, including the pre-registration of the study, the breadth of societies considered, the consideration of multiple norm violations and punishment strategies, and the effort to complement student samples with non-student samples when possible. That said, I think certain aspects of the paper would benefit from revision and/or elaboration, in particular, concerning: (1) the interpretation of findings, (2) the robustness of results across different norm violations, and (3) the level of detail provided in the methods and results sections. I'll elaborate on these points below and include a list of additional comments at the end.

(1) Interpretation of findings. My reading of the authors' proposition is that, when developing and establishing norms about punishment, societies need to make a trade-off between social harmony (which is better served by favoring indirect punishment) and effectiveness (which is better served by favoring direct punishment). The authors interpret their cross-cultural variability findings as supporting this position, based for example on the finding that societies with more individualistic values and looser norms tend to consider gossip as more appropriate, and physical and verbal confrontation as less appropriate, responses to offenses. This finding is interpreted as suggesting that these societies favor harmony perhaps at the expense of the effectiveness of punishment.

My main concern with this interpretation has to do with the question of where/how social ostracism fits in this picture. Conceptually, it seems that societies that would favor social harmony and conflict avoidance, would have meta-norms favoring various types of indirect punishment, including both gossip and ostracism. Indeed, this is part of the authors' expectations, as described in the pre-registration: "An alternative [hypothesis] is that there is complementarity between confronting and avoiding on the country-level, such that avoiding would be more appropriate when confronting is less appropriate." However, the results seem to point to a more complex pattern. (a) Looking at correlation results in Table 1, it is clear that the relations of cultural and environmental factors with confrontation and ostracism are of similar strength and direction, while their relations with gossip and non-action are typically in the opposite direction. (b) Relatedly, in Table S8, we see that meta-norms about physical confrontation are positively related with meta-norms about social ostracism, and that meta-norms about gossip are negatively related with meta-norms about social ostracism. How do these results fit with the proposed direct vs. indirect punishment distinction and the trade-off between effectiveness and conflict?

(2) Robustness across norm violations. My second question concerns the extent to which results are robust when considering different types of norm violations. In Table S4, the authors show that the appropriateness of punishment strategies was consistent across scenarios. Additionally, I would like to see whether the key results (pertaining to how cultural and environmental factors relate to punishment meta-norms) hold for different types of norm violations. In particular, it would be

interesting to see if the same patterns hold when considering only cooperation-relevant violations (the resource depletion animation) versus considering only the verbal scenarios (B-E), some of which seem rather mundane or merely quirky. This comparison could strengthen confidence that findings are robust across different methods of presenting norm violations, and across different domains of norm violations (cooperation-relevant or not). The pre-registration indeed includes this comparison, as well as a comparison between first-order and meta-scenarios, which I think would also be valuable to present.

(3) Completeness of presentation. My last major recommendation has to do with the level of detail in the presentation of methods and results. It would be helpful to provide some key additional information in the manuscript, including: an explanation of how the norm violation scenarios were chosen; an explicit explanation of where the 50 appropriateness ratings came from (10 violations + 4 punishment responses × 10 violations – this was clearer in the pre-registration); a more detailed description of results from Table 1 in text. Relevant to point (1) above, I would also suggest a more detailed interpretation of findings pertaining to social avoidance in the discussion.

Other comments:

- In lines 282- 286 of the manuscript, the authors describe previous work on the perception of punishers versus non-punishers, suggesting that non-punishers are not viewed more negatively. Here, it would be useful to compare to evidence that punishers receive substantial reputational benefits – e.g., see Barclay, 2006; Jordan et al., 2016; Patil et al., 2018.
- In the pre-registration, the authors mention running robustness checks excluding participants who failed the attention check, and those who found the study difficult to understand. Did they perform these checks and, if so, what were the results?
- It would be helpful to provide the syntax for the analyses (including standardization procedures, if possible).

References

- Barclay, P. (2006). Reputational benefits for altruistic punishment. *Evolution and Human Behavior*, 27(5), 325-344.
- Jordan, J. J., Hoffman, M., Bloom, P., & Rand, D. G. (2016). Third-party punishment as a costly signal of trustworthiness. *Nature*, 530(7591), 473-476.
- Patil, I., Dhaliwal, N. A., & Cushman, F. A. (2018). Reputational and cooperative benefits of third-party compensation. Available at <https://psyarxiv.com/c3bsj/>

Response to reviewers

We wish to thank all reviewers for their encouraging remarks and constructive criticisms, which has led to a substantial revision of the paper. As you will see, we address all issues -- large and small -- and the reviews also led us to follow the advice to follow the pre-registration more closely, especially with respect to hypotheses and analyses. In our view, these revisions yielded both a more compelling introduction, more focussed methods and results sections, as well as a more interesting and balanced discussion section.

Reviewer #1 (Remarks to the Author):

This cross-site cross-cultural study adds to our understanding of how societies maintain cooperation using norms about how to enforce norms. The work shows some interesting insights into how commonalities of normative influence are maintained across different groups as explained by cross-cultural dimensions of variation.

My major concern about this paper has to do with the way that the analyses are described in the main text. I find it hard to follow the explanation here as to how and why country means were used instead of retaining the full information of the data from individual respondents. In the method section, some but not all of the means used in the analyses are explained as derived from multilevel models and (might?) be empirical Bayes means. I see references to the preregistration, but I think it would do to repeat some of the key methods here at least in the supplement as the preregistration documentation could become hard to follow up as this publication gets older.

RESPONSE: Great point. The introduction now explicitly lays out the pre-registered hypotheses, from which it should be clear that they specifically concern country-level variation of metanorms. Moreover, the analyses, which were also pre-registered, should be easier to follow in this revision. Calculation of metanorms is now discussed in the Results section under the subheading "Country measures of metanorms" (l. 344-355) with details in the Methods section under the subheading "Calculation of metanorm measures" (l. 618-635).

As a similar but more minor point, the axes in figure 1 are unclear and it would be much easier to read this plot with labels on the axes.

RESPONSE: Agreed. We have added labels added to Fig. 1, which now looks as follows;

Figure 1. Within-country correlations between norm violation appropriateness and response appropriateness. The vertical axis refers to the correlation, across scenarios, between the (country-level) appropriateness ratings of norm violations and a given response to them. The boxplots show that, in essentially every country, verbal confrontation, gossip, and social ostracism all yielded negative correlations, while non-action yielded positive correlations. The dashed reference line indicates a zero correlation. The box represents the interquartile range with the dark line indicating the median. The whiskers reach the min and max values in case these are at most 1.5 times the box height outside the interquartile range, otherwise outliers are marked by circles.

I also find the method detail on when and how 40-year-old aggregate data on cultural dimensions were used vs. the new measures of these dimensions from the data collected for this study. It is unclear to me which countries had new dimensional data collected, which were reliant on the old data, and which were imputed. Perhaps a table added to the supplement would help.

RESPONSE: We agree that the hybrid measures of culture were difficult to understand, but they have been discarded altogether in this version. Instead, in this revision, we follow more explicitly the preregistration by using only our survey data to obtain culture measures. However, we make an exception for the cultural dimensions of indulgence, power distance, and individualism, because a check of the internal consistency of these scales indicated that they appeared to have very low reliability. Thus for these cultural dimensions we deviate from the preregistration and use the original Hofstede measures instead. This is now explained in the Methods section, under the subheadings “Culture measures” (l. 638-640) and “Changes to the pre-registered analyses” (l. 683-686).

I also think it would also be informative to add some kind of figure like a map showing the relative proportion of the sample collected from different regions of the world. There clearly are some non-WEIRD populations included, though all appear to be from either student and/or urban centres. Having some indication of how widely the cross-cultural sample expands beyond the usual Western world would be helpful.

RESPONSE: In addition to the color-coded maps in Fig. 2 (which show which regions of the world were sampled), we have added a sentence to the introduction (l. 222-223) saying how many countries were sampled from each continent: “To compare the appropriateness of different forms of sanctions across societies we conducted a survey study in 57 countries across the globe, including 7 African countries, 10 American countries, 18 Asian countries, 21 European countries, and Australia.”

Reviewer #2 (Remarks to the Author):

This is a very interested and well-conducted paper. The authors examine how people respond to rule-breakers across societies, and they show how different types of reactions are predicted by country-level factors such as median income, individualistic values, and gender equality. This is a topic of great theoretical importance: in addition to being a cross-cultural comparison of norms and the socio-ecological causes thereof (which is novel unto itself), it also speaks towards the literature on punishment and what maintains certain norms in different societies. This is an ambitious project, with data from an impressive 57 countries, and at least 88 co-authors. Altogether, this is a great piece of work and I think it is highly appropriate for Nature Communications and will be well-cited. I have some recommendations for minor revisions, but the authors should be able to fix these upon revision.

My two biggest comments are as follows:

1. First, many countries rate the most appropriate action as “non-action”. This warrants discussion. What does this say about whether there is indeed a meta-norm? The results are interesting either way, but the authors might need to tone down whether or not the meta-norm does or does not support the norm.

RESPONSE: This is a very interesting comment. In the Discussion (l. 450-458) we now address it as follows: “Our study also contributed to the longstanding debate on whether metanorms require punishment of norm violators. Theoretical work on altruistic punishment has often assumed that not punishing a norm violation is selfish and hence should be deemed inappropriate²⁸, and studies using economic experiments have found that those who pay a cost to punish others’ selfish behavior are subsequently trusted more than non-punishers^{29,30}. However, studies have also found that non-punishers are not viewed as more selfish^{27,31} and do not elicit more disapproval^{13,17,18}. Our finding that non-action was the most appropriate response in many countries is consistent with this latter research, supporting the notion that metanorms often do not require bystanders to punish norm violators.”

2. Second, I worry that the authors are extrapolating too much from the proportion of variance explained by country. This is just a small point in main text (an analysis of a secondary or tertiary point), but it is relevant to some larger debates, and it may be extrapolating farther than it should. First, the analysis across cities only includes student samples. We should expect some similarity in student samples within a country, not only because there may be a common “student culture”, but also because there is often mixing of students from different regions, i.e., students at a given university may come from many regions, which decreases the variance among “cities” within a country. Second, the comparison of student & non-student samples is all from people residing in cities. In many countries, the culture is different within cities than it is in rural areas. For example, in the USA, the Republican vs. Democrat divide is less of a state-by-state difference than it is a rural vs. urban difference. A similar pattern holds in Canada, with Conservatives dominating outside of urban centres. Similarly, analyses of Ultimatum Games show that while participants in industrialized countries behave similarly by making & demanding fair offers, participants in small-scale societies often behave differently by making & tolerating less fair offers (see Joe Henrich’s work). Other work has shown very different behaviours in different parts of the same city, whereby affluent areas are more cooperative (for example) than less affluent areas (see Daniel Nettle’s work in Newcastle UK; I believe David Sloan Wilson may have similar patterns in Binghamton NY). Thus, depending on how the researchers in each country sampled the non-student populations, it might have resulted in more homogeneity if the non-student samples were collected from more affluent parts of town surrounding the universities. This will reduce the variance within countries, making the country-level variance appear higher than it actually is. If the authors wish to make strong claims about the proportion of variance attributable to country vs. city or student-non-student, then we need more information on how each sample recruited non-student participants. Otherwise, the authors must present significant cautions around the interpretation of such results, for example a summary of the above points. These cautions must be in the main text, though it could just be a short summary with more elaboration in the Supplement next to Table S5. I should note that it’s good that the authors only compared the 10 countries that had >1 city, otherwise the variance accounted by country would have

been inflated by the fact that only 10/57 countries had >1 country. Same with only comparing the 31 countries with student + non-student samples.

RESPONSE: We agree that it is an open question whether our country scores are representative for rural and less affluent parts of the countries. We have revised the relevant part of the Discussion (l. 529-534) as follows: "Finally, our sampling strategy had both strengths and limitations. By collecting data from both students and non-students, and across different cities, we established that these subsamples tended to have similar metanorms in the same country. However, it is possible that metanorms exhibit within-country variation along the urban-rural and affluent-poor dimensions, which we were unable to capture when focusing on urban locations with universities."

3. Third, the authors should clarify if they get the same results if standardizing with the rating for non-action instead of average of all items. I believe this is done somewhere, and if so, it should come out clearer in the writing, preferably early (i.e., when describing standardizing based on averages).

RESPONSE: We have added the following passage in the Methods section under the subheading "Unregistered analyses" (l. 700-706): "Standardizing by the metanorm for non-action. Subtraction of the metanorm for non-action from the metanorms for sanctions yields a measure of how appropriate the sanction is relative to doing nothing at all. However, this method has the drawback that ratings for non-action exhibit meaningful country variation (as seen in Table 2), which will be incorporated in the measures for every sanction, thereby making them artificially more closely intercorrelated. Nonetheless, the pattern of results for how metanorms vary across cultures remains qualitatively the same (see Supplementary Table 8)."

The following are all very minor comments:

- Figure 1 should have some title on y axis
- Specify boxes and whiskers in figure 1
- When discussing the lack of global consensus, the authors should specify that some countries had no sanction as most appropriate action
- Line 502 (Methods) says the controlled metanorm estimates are in Table S4, but it's actually S3
- Table S7 is potentially very useful for other researchers. To facilitate meta-analyses, the authors should include 95% CI's or something similar to give an estimate of the uncertainty. (This should probably be included anyway, but will be especially useful for future meta-analyses.)
- On lines 284-286, the authors claim that non-punishers are not viewed harshly in other literature. While this is strictly true that non-punishers are not punished, multiple studies going back a couple decades show that non-punishers are trusted less than are punishers. These include:
Barclay, P. (2006). Reputational benefits for altruistic punishment. *Evolution and Human Behavior*, 27, 325-344.

Nelissen, R. (2008). The price you pay: cost-dependent reputation effects of altruistic punishment. *Evolution & Human Behavior*, 29(4), 242-248.

Raihani, N.J., & Bshary, R. (2015). Third-party punishers are rewarded, but third-party helpers even more so. *Evolution*, 69, 993-1003.

Jordan, J.J., Hoffman, M., Bloom, P., & Rand, D.G. (2016). Third-party punishment as a costly signal of trustworthiness. *Nature*, 530, 473-476.

See also: Raihani, N.J., & Bshary, R. (2014). The reputation of punishers. *Trends in Ecology & Evolution*, 30(2), 98-103.

This work should be mentioned.

RESPONSE: We have cited two of the suggested papers (Barclay and Jordan et al.) as examples of this line of work, and we have made the other suggested edits, with the sole exception of including CIs in Supplementary Table 7. Unfortunately, the table will not fit on the width of a page if CIs are included. For the purpose of the present paper, the reported intercorrelations of the predictor variables are just descriptive and not part of any hypotheses about how they correlate outside our sample of countries; moreover, our data are open at an OSF site. Thus, any researcher interested in these CIs can easily calculate them.

I think that the authors can address these comments upon review. If they can do so, then I think that this paper will make a strong addition to the literature. I am happy to elaborate on any of my comments if the authors wish.

Signed,

Pat Barclay, University of Guelph, barclayp@uoguelph.ca

Reviewer #3 (Remarks to the Author):

This paper is an impressive undertaking. The authors ran a preregistered study on the appropriateness of different responses to norm violations in 57 countries, translating the survey into 30 languages and, when possible, running it with both student and non-student samples. The authors find that people in different societies vary in how much they endorse different responses to norm violations and that these differences correlate, to some extent, with ecological, institutional, and cultural variables. The data are important and relevant to ongoing debates in the literature on norm enforcement, cooperation, and cross-cultural variation. But, as currently written, the paper suffers from several shortcomings:

The biggest limitation of the manuscript is that it isn't clear what the theoretical stakes are. The authors write that previous work has shown that norms governing punishment vary with individualism and tightness and that these are linked to other institutional and ecological factors. They go on to show that physical sanctions exhibit these patterns, verbal sanctions less so. That's fine, of course, but it's not clear why these results matter.

How should the reader interpret the fact that tightness, etc. seem to affect physical sanctioning but not verbal sanctioning?

I recommend that the authors beef up the theoretical presentation in the introduction. To say that they make predictions based on results in previous papers is fine, but it would be much more compelling and important if they grounded this in larger theory. Perhaps the theory is that, for whatever reason (cultural group selection? Adaptive psychology?), under conditions of greater social and ecological threat, societies enforce norms more tightly (to promote coordination?). Perhaps this theoretical grounding makes a suite of predictions that the authors can precisely specify (such as in a table). Perhaps there are reasons to expect that gossip, physical sanctioning, ostracism, and verbal sanctioning should all respond similarly to tightness and social ecological threat; perhaps there are reasons not to expect similarities. Regardless, the paper would benefit from being clearer on which theory is being tested and what the predictions of that theory are. This would also mean editing the discussion, pointing out which results were predicted, which results violated predictions, and what that might mean. (In that vein, it now reads like the gossip results were surprising, but it's not clear whether the "hypotheses" made predictions about gossip, or whether those are more exploratory.)

Related to the last point, it wasn't clear to what extent the predicted relationship between tightness/threat, on the one hand, and appropriateness of norm enforcement, on the other, was supported. Tightness was positively correlated with the acceptance of physical confrontation yet seemed uncorrelated with meta-norms about verbal sanctioning; the same was true for gender equality. Perceived threat seemed uncorrelated with metanorms for both physical sanctioning and verbal sanctioning. This seems either inconvenient for the authors or theoretically interesting (or both). Regardless, it should be discussed.

EDIT: I see that the preregistration presents clear hypotheses with theoretically rooted justifications and precise predictions. These should be included in the paper. If the authors choose a set of hypotheses or predictions aside from those in the pre-registration, or if they choose not to present some of them, then that divergence should be justified.

RESPONSE: Excellent points. This very valuable advice encouraged us to revise the whole manuscript to closely follow the preregistration, thereby providing five detailed hypotheses in the theory section (l. 238-326), which are then clearly evaluated in the results section (l. 328-453). The discussion has also been edited and structured to follow the series of hypotheses. The small divergences from the pre-registration are summarized in the Methods section under subheadings "Changes to the preregistered analyses" and "Unregistered analyses" (l. 679-706).

It seemed odd that gossip was presented as a sanction. People don't seem to gossip as a way of incentivizing good behavior; they seem to gossip to share and acquire information. The fact that gossip incentivizes norm compliance seems incidental. (Relatedly, the idea expressed in lines 250-252 that a correlation between the appropriateness of some behavior X and the inappropriateness of a norm violation shows that X is a sanction seemed odd, given that other behaviors (such as a desire to acquire information) seem like they would show a similar pattern.)

RESPONSE: We now elaborate on what we mean by informal sanctions in the introduction (l. 201-220): “Existing research often examines and conceptualizes sanctions in generic terms as a form of punishment that reduces outcomes for another person.^{7,8} While parsimonious, this characterization is unlikely to provide a realistic account of how people deal with norm violators in everyday life. To capture this realism, a few scholars have recently proposed three distinct informal sanctions:^{9,10} social ostracism (e.g., individuals or groups actively avoiding someone), gossip (e.g., spreading information about someone’s inappropriate behavior), and direct confrontation (e.g., verbal or physical). Although these responses may not always be intended to modify the norm violator’s behavior, they can all be viewed as expressions of disapproval that serve to strengthen a given norm. According to these theoretical perspectives, social ostracism, gossip, and direct confrontation are sanctions, even though the sanctioned party is not necessarily aware of being sanctioned. Indeed, this may be a key reason that potential norm enforcers prefer one response over another. For instance, whereas direct confrontation should be especially effective at making the norm violator aware of why they are being sanctioned and thus change their behavior, gossip should be less likely to evoke direct conflict but can still promote norm compliance by making the norm more salient in the group. Similarly, physical confrontation may be harmful in a way that verbal confrontation is not. And social ostracism may directly harm targets’ opportunities whereas gossip may harm targets more indirectly via reputational damage. Given these differences between sanctions, it is noteworthy that prior cross-cultural work has rarely distinguished between forms of sanctions, instead focusing on costly actions that reduce outcomes for another person in economic games¹¹⁻¹², physical confrontation,¹³ or unspecified ‘punishment’¹⁴.”

Physical sanction is often excluded from comparisons among the other reactions to norm violations (e.g., in Figure 1). Why is this? If it’s because the physical sanctioning data could not be compared to the other data because of how they were collected, that might be worth saying early in the paper (although it isn’t clear why they were comparable for Figure 2 but not comparable for Figure 1). Without addressing early in the paper why physical sanctioning data are only sometimes presented, readers may suspect that the authors are intentionally excluding data.

RESPONSE: Physical sanctioning data are based only on two scenarios instead of ten, and this is the reason they are not exactly comparable to the other sanctioning data; e.g., Figure 1 presents correlations of appropriateness ratings of norm violations (across the scenarios) with appropriateness ratings of four responses (verbal confrontation, social ostracism, gossip, non-action). Such correlations could not be calculated for physical sanctioning. The special status of physical sanctioning data is now made clearer in the Results section under the subheading “Country measure of metanorms” (l. 344-355): “Following our preregistration, we calculated country measures of metanorms for each of four responses (verbal confrontation, social ostracism, gossip, non-action) by using country-mean appropriateness ratings for the five scenarios in the non-cooperation and out-of-place behavior domains. These were adjusted for variation in ratings of the

appropriateness of the underlying norm violations (see Methods). In the preregistration we assumed that metanorms for verbal and physical confrontation would be the same, but recent work has shown that they may be viewed quite differently.¹⁰ We therefore separately calculated the country measures of metanorms for physical confrontation by averaging the country-mean appropriateness ratings of two meta-violation scenarios in which physical confrontation was used in two different contexts: against an agent depleting a common resource and against someone insulting a man's mother."

Smaller comments:

Regarding this statement, "Prior cross-cultural work has not distinguished between forms of sanctions", I was surprised to read that. If it's true, that adds to the importance of the paper. I was quickly reminded of this pre-print by Moya et al., <https://drive.google.com/file/d/1tIKFkScdggCvgsu-Qr7XgGKur-SA7jW/view>, although the authors may not count it for some reason (e.g., it's still in the review process). Regardless, some readers may find this sentence surprising, so it seems worth explaining it in greater depth. The authors cited papers by Brauer & Chaurand, Eriksson et al., and Gelfand et al., but it would help to be clear on whether other cross-cultural projects on norms and norm enforcement (e.g., by House et al., Henrich et al., Barrett et al., Schulz) also fail to distinguish between forms of sanctions.

RESPONSE: We have added references to House et, Henrich et al. and Barrett et al. in the revised sentence (l. 217-220): "Given these differences between sanctions, it is noteworthy that prior cross-cultural work has rarely distinguished between forms of sanctions, instead focusing on costly actions that reduce outcomes for another person in economic games¹¹⁻¹², physical confrontation,¹³ or unspecified 'punishment'¹⁴."

This sentence seemed unjustified: "Finally, by collecting data from both students and nonstudents, and across different locations in the same country, we established that metanorms look pretty much the same in different groups within a society, even though they differ across societies." It's important and valuable that the authors collected non-student samples, but this was done in a subset of the countries and the non-student samples looked like they often (always?) came from the same urban areas as the student samples (in two cases, it sounded like, they were collected at universities). Surveying English-speaking students and non-students in Mumbai, for instance, doesn't seem like it shows that metanorms look pretty much the same across India.

RESPONSE: The same point was raised by Reviewer 2. We have revised the section in the discussion (l. 529-534) as follows: "Finally, our sampling strategy had both strengths and limitations. By collecting data from both students and non-students, and across different cities, we established that these subsamples tended to have similar metanorms in the same country. However, it is possible that metanorms exhibit within-country variation along the urban-rural and affluent-poor dimensions, which we were unable to capture when focusing on urban locations with universities."

Reviewer #4 (Remarks to the Author):

This manuscript describes an impressive study of meta-norms about peer punishment across a diverse set of 57 societies, with a large number of participants (N = 22,863). It examines how the appropriateness of norm violations and distinct responses to norm violations (confrontation, gossip, and ostracism) varies across cultures, and how cultural (e.g., individualism, tightness) and environmental (e.g., pathogen stress, threat estimates) factors relate to norms about more vs. less confrontational punishment of rule-breakers.

The authors provide compelling evidence for cultural universals and cultural variation in meta-norms. They find that, across societies, people view the use of verbal confrontation, gossip, and ostracism as more acceptable, the more inappropriate a norm violation. At the same time, they observe substantial cross-cultural variation in the types of punishment that are deemed most appropriate: while some societies favor confrontational strategies, others favor indirect strategies of gossip or ostracism. Importantly, cultural and environmental factors are used to explain this variation. As a case in point, median per-capita income and gender gap indicators differentially relate to meta-norms about confrontation versus gossip. In countries with higher median income and gender equality, the acceptability of confrontation is lower while the acceptability of gossip is higher.

In my view, these results offer novel insights, pertinent to our understanding of the evolution of cooperation, the functions of punishment strategies, and cultural universality and variability. Findings underscore the importance of studying norms about distinct types of punishment (moving beyond a current focus only on costly, direct punishment). They provide support for the exciting possibility that different societies find different solutions – by developing norms that favor distinct punishment strategies – to promote cooperation and deter rule-breakers. And they move one step further to examine which societal-level factors relate to meta-norms about distinct forms of punishment. These findings have the potential to be highly influential among researchers in different fields—evolutionary biology, cultural evolution, psychology, and behavioral economics.

Going into the nuts and bolts of the paper, there are multiple aspects to appreciate, including the pre-registration of the study, the breadth of societies considered, the consideration of multiple norm violations and punishment strategies, and the effort to complement student samples with non-student samples when possible. That said, I think certain aspects of the paper would benefit from revision and/or elaboration, in particular, concerning: (1) the interpretation of findings, (2) the robustness of results across different norm violations, and (3) the level of detail provided in the methods and results sections. I'll elaborate on these points below and include a list of additional comments at the end.

(1) Interpretation of findings. My reading of the authors' proposition is that, when developing and establishing norms about punishment, societies need to make a trade-off between social harmony (which is better served by favoring indirect punishment) and effectiveness (which is better served by favoring direct punishment). The authors interpret their cross-cultural variability findings as supporting this position, based for example on

the finding that societies with more individualistic values and looser norms tend to consider gossip as more appropriate, and physical and verbal confrontation as less appropriate, responses to offenses. This finding is interpreted as suggesting that these societies favor harmony perhaps at the expense of the effectiveness of punishment. My main concern with this interpretation has to do with the question of where/how social ostracism fits in this picture. Conceptually, it seems that societies that would favor social harmony and conflict avoidance, would have meta-norms favoring various types of indirect punishment, including both gossip and ostracism. Indeed, this is part of the authors' expectations, as described in the pre-registration: "An alternative [hypothesis] is that there is complementarity between confronting and avoiding on the country-level, such that avoiding would be more appropriate when confronting is less appropriate." However, the results seem to point to a more complex pattern. (a) Looking at correlation results in Table 1, it is clear that the relations of cultural and environmental factors with confrontation and ostracism are of similar strength and direction, while their relations with gossip and non-action are typically in the opposite direction. (b) Relatedly, in Table S8, we see that meta-norms about physical confrontation are positively related with meta-norms about social ostracism, and that meta-norms about gossip are negatively related with meta-norms about social ostracism. How do these results fit with the proposed direct vs. indirect punishment distinction and the trade-off between effectiveness and conflict?

RESPONSE: We also found the pattern of results for social ostracism vs gossip surprising. We now devote more attention to this important point in the Discussion section (l. 460-472): "When designing the study to include several forms of informal sanctions, it was an open question to us whether different forms would exhibit similar cross-cultural variation in appropriateness. We speculated that there could instead be complementarity between preferences for confrontation and preferences for non-confrontational sanctions, such as social ostracism and gossip. Indeed, we did find a clear fault line between societies condoning physical confrontation and societies condoning gossip—but, surprisingly, metanorms for gossip were negatively correlated with metanorms for social ostracism, which instead were positively correlated with metanorms for physical confrontation. It was also surprising that metanorms for physical and verbal confrontation were only weakly correlated. These results indicate that metanorms are sanction-specific. This interpretation was supported by the additional finding that metanorm measures for distinct sanctions correlated well with the reported levels of use of the same sanctions. Nonetheless, the observed pattern of consistencies and complementarities across different forms of informal sanctions remain intriguing puzzles that require further research. We offer some thoughts below. "

Later in the Discussion (l. 496-503) we return to this point: "What is it about gossip that makes its appropriateness change in ways distinct from social ostracism, which is another non-confrontational sanction? One key difference is whether the response is directed to the norm-violator or a third party. Specifically, confrontation and active avoidance concern responses that are related to how you interact with the norm-violator; in contrast, gossiping concerns how you interact with another person. For this reason, people may

think of both confrontation and social ostracism as ‘punishment’ while viewing gossip in a different way, even though they are all expressions of disapproval and even though gossip may be as effective in sustaining norms.³³

(2) Robustness across norm violations. My second question concerns the extent to which results are robust when considering different types of norm violations. In Table S4, the authors show that the appropriateness of punishment strategies was consistent across scenarios. Additionally, I would like to see whether the key results (pertaining to how cultural and environmental factors relate to punishment meta-norms) hold for different types of norm violations. In particular, it would be interesting to see if the same patterns hold when considering only cooperation-relevant violations (the resource depletion animation) versus considering only the verbal scenarios (B-E), some of which seem rather mundane or merely quirky. This comparison could strengthen confidence that findings are robust across different methods of presenting norm violations, and across different domains of norm violations (cooperation-relevant or not). The pre-registration indeed includes this comparison, as well as a comparison between first-order and meta-scenarios, which I think would also be valuable to present.

RESPONSE: As one would expect from lower measurement reliability, results become somewhat weaker when measures are based on only the cooperation scenario rather than the aggregate of scenarios; but the direction of results remains the same. The results are reported in Supplementary Table 6. In the Results section, under subheading “Hypothesis 5”, we have now added (l. 424-425): “These correlations also tend to keep the same signs when metanorms are estimated separately for non-cooperation and out-of-place behaviors, see Supplementary Table 6.”

(3) Completeness of presentation. My last major recommendation has to do with the level of detail in the presentation of methods and results. It would be helpful to provide some key additional information in the manuscript, including: an explanation of how the norm violation scenarios were chosen;

RESPONSE: In the Methods section, under subheading Scenarios, we explain how the norm violation scenarios were chosen (l. 579-595): “Scenarios were selected to cover potentially inappropriate behavior in three domains: cooperation, out-of-place everyday behavior, and meta-violations (i.e., potentially inappropriate use of an informal sanction). The cooperative domain was covered by an animation of an agent depleting a common resource, referred to as scenario A. This scenario was drawn from prior research on metanorms.¹³ Out-of-place everyday behavior was covered by four scenarios describing someone (B) listening to music on headphones at a funeral, (C) sleeping in a restaurant, (D) singing in a library, or (E) reading a newspaper at the movies. These combinations of behaviors and contexts were found to be widely viewed as inappropriate in a prior cross-cultural study of norms.¹⁵ Meta-violations included two instances of physical confrontation: (F) an animation of an agent physically confronting someone who depleted a common resource in scenario A, and (G) a verbal scenario with a man being physically aggressive against someone who insulted his mother. We use scenarios F and G to

calculate metanorm measures for physical confrontation. The remaining three meta-violation scenarios described someone who reacted to a person who was rude in a public place in one of three ways: (H) by reprimanding this person, (I) by speaking negatively about this person, or (J) by staying away from this person.”

an explicit explanation of where the 50 appropriateness ratings came from (10 violations + 4 punishment responses × 10 violations – this was clearer in the pre-registration);

RESPONSE: In the introduction we now write (l. 231-236): “For each of the ten scenarios, participants rated the appropriateness of the described behavior as well as the appropriateness of four different responses to it: verbal confrontation (making an angry remark to the norm violator), gossip (talking to someone else about the norm violator), social ostracism (making a point of avoiding the norm violator in the future), and non-action (doing nothing), for a total of 10 × 5 = 50 ratings.”

a more detailed description of results from Table 1 in text.

RESPONSE: This table is now referred to as Table 2. In the Results section, under subheading “Hypothesis 5”, we now write (l. 414-425): “As preregistered, we calculated the pairwise correlations between the various country measures and our metanorm measures (Table 2). For physical confrontation, all correlations showed the predicted direction. Particularly strong results ($r > .50$) were obtained for power distance, individualism, individual autonomy, emancipative moral judgments, tightness, gender equality, and median income. Results for verbal confrontation were weaker and two of the correlations (for tightness and pro-violence attitudes) went weakly in the wrong direction. Thus, Hypothesis 5 received support but much more strongly for physical confrontation than for verbal confrontation, underscoring the need for making a distinction between these sanctions. Consistent with our previous analysis of sanction-specificity of metanorms, results for social ostracism showed the same pattern as for physical confrontation, whereas results for gossip followed the opposite pattern. (These correlations also tend to keep the same signs when metanorms are estimated separately for non-cooperation and out-of-place behaviors, see Supplementary Table 6.)”

Relevant to point (1) above, I would also suggest a more detailed interpretation of findings pertaining to social avoidance in the discussion.

RESPONSE: See our response to point (1) above.

Other comments:

- In lines 282- 286 of the manuscript, the authors describe previous work on the perception of punishers versus non-punishers, suggesting that non-punishers are not viewed more negatively. Here, it would be useful to compare to evidence that punishers receive substantial reputational benefits – e.g., see Barclay, 2006; Jordan et al., 2016; Patil et al., 2018.

RESPONSE: Reviewer 2 made the same point. We have cited two of the suggested papers (Barclay and Jordan et al.) as examples of this line of work (l. 452-454).

- In the pre-registration, the authors mention running robustness checks excluding participants who failed the attention check, and those who found the study difficult to understand. Did they perform these checks and, if so, what were the results?

RESPONSE: Results are indeed robust, because metanorms measures with and without exclusions correlate at $r = .99$. We now report this in Supplementary Table 4. In the Results section, under the subheading “Country measures of metanorms”, we write (l.370-371): “Metanorm measures were virtually unchanged in analyses that excluded participants who failed attention or comprehension checks (Supplementary Table 4).”

- It would be helpful to provide the syntax for the analyses (including standardization procedures, if possible).

RESPONSE: A file SPSS Syntax.docx describing the syntax for the various analyses is now added to the OSF site where the data are deposited.

References

- Barclay, P. (2006). Reputational benefits for altruistic punishment. *Evolution and Human Behavior*, 27(5), 325-344.
- Jordan, J. J., Hoffman, M., Bloom, P., & Rand, D. G. (2016). Third-party punishment as a costly signal of trustworthiness. *Nature*, 530(7591), 473-476.
- Patil, I., Dhaliwal, N. A., & Cushman, F. A. (2018). Reputational and cooperative benefits of third-party compensation. Available at <https://psyarxiv.com/c3bsj/>

REVIEWERS' COMMENTS

Reviewer #1 (Remarks to the Author):

The authors have undertaken a major revision of the manuscript following reviewer feedback. The paper now much more closely aligns with the preregistration and is overall much clearer. I believe that, with these revisions, the paper is now ready for publication.

Reviewer #2 (Remarks to the Author):

This manuscript is improved over the original, and the authors have addressed all my concerns satisfactorily. I have am now happy to recommend publication.

I have one final minor thought. According to the authors, the raw numbers suggest that non-action was viewed as the most appropriate response in many countries. However, this is likely because the norm violations were all relatively minor. The main point of Figure 1 is that when violations become more serious (i.e., less appropriate), sanctions become viewed as more appropriate and non-action is viewed as less appropriate. Thus, although non-action was viewed more positively than sanctions in this study, this pattern probably only holds for minor norm violations (e.g., singing in library), and is unlikely to hold for serious infractions. This is worth mentioning whenever the authors mention that non-actions seemed to be preferred. For example, in the paragraph on lines 459-467, or lines 368-369, the authors should add some caveat like "Although non-action was deemed more appropriate than sanctions in many countries, this may be because all our norm violations were relatively minor, whereas Figure 1 suggests we should predict the opposite for more serious infractions." (or something similar)

Other than that minor caveat, I think this paper is ready to publish. Signed, Pat Barclay

Reviewer #3 (Remarks to the Author):

The manuscript is much improved following the revisions. I have only one final recommendation, which is that the authors discuss the gossip results in greater detail in the discussion fo results for Hypothesis 5 (see especially lines 420-432). Currently, that section focuses almost exclusively on physical confrontation and verbal confrontation.

Reviewer #4 (Remarks to the Author):

As mentioned in my previous review, I think this is an impressive, well-conducted cross-cultural study with important theoretical implications and intriguing empirical findings. I think the paper makes a substantial contribution to research on cooperation, norm enforcement and cross-cultural variability. I am satisfied with how the authors addressed the concerns raised in my previous review. And I think the revised version of the manuscript is improved by the inclusion of the pre-registered hypotheses, a more detailed presentation of methods and findings pertaining to each hypothesis, and an elaborate discussion of theoretical implications of this work. I especially appreciate the authors' thoughts on the observed patterns for physical confrontation and ostracism versus gossip, as well as the discussion of why gossip may be a special type of sanction.

RESPONSE TO REVIEWERS

Reviewer #1 (Remarks to the Author):

The authors have undertaken a major revision of the manuscript following reviewer feedback. The paper now much more closely aligns with the preregistration and is overall much clearer. I believe that, with these revisions, the paper is now ready for publication.

RESPONSE: Thank you!

Reviewer #2 (Remarks to the Author):

This manuscript is improved over the original, and the authors have addressed all my concerns satisfactorily. I am now happy to recommend publication.

I have one final minor thought. According to the authors, the raw numbers suggest that non-action was viewed as the most appropriate response in many countries. However, this is likely because the norm violations were all relatively minor. The main point of Figure 1 is that when violations become more serious (i.e., less appropriate), sanctions become viewed as more appropriate and non-action is viewed as less appropriate. Thus, although non-action was viewed more positively than sanctions in this study, this pattern probably only holds for minor norm violations (e.g., singing in library), and is unlikely to hold for serious infractions. This is worth mentioning whenever the authors mention that non-actions seemed to be preferred. For example, in the paragraph on lines 459-467, or lines 368-369, the authors should add some caveat like "Although non-action was deemed more appropriate than sanctions in many countries, this may be because all our norm violations were relatively minor, whereas Figure 1 suggests we should predict the opposite for more serious infractions." (or something similar)

Other than that minor caveat, I think this paper is ready to publish. Signed, Pat Barclay

RESPONSE: Thank you, we appreciate this suggestion. In the discussion, after the sentence "Our finding that non-action was the most appropriate response in many countries is consistent with this latter research, supporting the notion that metanorms often do not require bystanders to punish norm violators", we have added the requested caveat as follows: "Note that this conclusion applies only to relatively minor norm violations, as the appropriateness of non-action was found to decrease for more serious infractions (Fig. 1)."

Reviewer #3 (Remarks to the Author):

The manuscript is much improved following the revisions. I have only one final

recommendation, which is that the authors discuss the gossip results in greater detail in the discussion fo results for Hypothesis 5 (see especially lines 420-432). Currently, that section focuses almost exclusively on physical confrontation and verbal confrontation.

RESPONSE: We appreciate this suggestion and have extended the relevant paragraph as follows: "Results for social ostracism showed the same pattern as for physical confrontation. However, results for gossip followed the exact opposite pattern. For example, the appropriateness of gossip was higher in countries that were *higher* on individualism, autonomy values, emancipative moral judgments, gender equality, and median income. This opposite pattern for gossip is consistent with our previous analysis of sanction-specificity of metanorms."

Reviewer #4 (Remarks to the Author):

As mentioned in my previous review, I think this is an impressive, well-conducted cross-cultural study with important theoretical implications and intriguing empirical findings. I think the paper makes a substantial contribution to research on cooperation, norm enforcement and cross-cultural variability. I am satisfied with how the authors addressed the concerns raised in my previous review. And I think the revised version of the manuscript is improved by the inclusion of the pre-registered hypotheses, a more detailed presentation of methods and findings pertaining to each hypothesis, and an elaborate discussion of theoretical implications of this work. I especially appreciate the authors' thoughts on the observed patterns for physical confrontation and ostracism versus gossip, as well as the discussion of why gossip may be a special type of sanction.

RESPONSE: Thank you!